# An ultrathin conformable vibration-responsive electronic skin for quantitative vocal recognition

Siyoung Lee[1], Junsoo Kim[2], Inyeol Yun [3], Geun Yeol Bae[1], Daegun Kim[1], Sangsik Park[1], Il-Min Yi[3], Wonkyu Moon[2], Yoonyoung Chung [3] & Kilwon Cho[1]

Flexible and skin-attachable vibration sensors have been studied for use as wearable voice-recognition electronics. However, the development of vibration sensors to recognize the human voice accurately with a flat frequency response, a high sensitivity, and a flexible/conformable form factor has proved a major challenge. Here, we present an ultrathin, conformable, and vibration-responsive electronic skin that detects skin acceleration, which is highly and linearly correlated with voice pressure. This device consists of a crosslinked ultrathin polymer film and a hole-patterned diaphragm structure, and senses voices quantitatively with an outstanding sensitivity of $5.5\,V\,Pa^{-1}$ over the voice frequency range. Moreover, this ultrathin device ($<5\,\mu m$) exhibits superior skin conformity, which enables exact voice recognition because it eliminates vibrational distortion on rough and curved skin surfaces. Our device is suitable for several promising voice-recognition applications, such as security authentication, remote control systems and vocal healthcare.

[1] Department of Chemical Engineering, Pohang University of Science and Technology, Pohang 37673, Korea. [2] Department of Mechanical Engineering, Pohang University of Science and Technology, Pohang 37673, Korea. [3] Department of Electrical Engineering, Pohang University of Science and Technology, Pohang 37673, Korea. Correspondence and requests for materials should be addressed to Y.C. (email: ychung@postech.ac.kr) or to K.C. (email: kwcho@postech.ac.kr)

The human voice is the most valuable biosignal for communication, and is essential not only in telecommunication, but also in human-machine interaction (HMI) and the Internet of Things (IoT), such as in remote control systems, home automation, and smart industrial infrastructure[1,2]. Microphones have been developed to detect human voice accurately, and applied in various electronic devices. Especially capacitive microphones, which typically exhibit higher sensitivity and lower noise level than other types of sensors[3], have been widely studied. The performance was steadily enhanced by adopting new electret materials[4] and a variety of structures such as corrugated diaphragm[5], hole-patterned backplate[6], and dual backplate[7], which results in a successful commercialization. Recently, flexible and skin-attachable sensors[8–20] that detect human voices by measuring the vibrations in users' neck skin have been researched. Such sensors have the unique advantages of comfortable fit and clear voice detection even in noisy or windy environments when compared to conventional rigid microphones[8]. Previous studies have reported various mechanisms to increase their sensitivity and accuracy: triboelectric[9–11]/piezoelectric[12–14,21]/piezocapacitive[15,16] and piezoresistive[8,17–20] sensors. However, previously reported skin-attachable sensors did not satisfy several essential requirements of microphone: flat frequency response, high sensitivity, and linearity of sensitivity. Those sensors did not maintain uniform sensitivity over the voice frequency range due to mechanical resonance and damping effects[9–11,17,22]. Especially, the sensors made of viscoelastic polymers[10–13,17–19] involving beta transition such as movements of side groups and long molecular chains[23], exhibited considerable damping effects. In addition, such sensors are not capable of measuring a voice pressure accurately[8,12], because a quantitative correlation between neck skin vibration and voice pressure was not fully understood. In addition, such sensors exhibit insufficient and non-linear sensitivity to distinguish subtle differences in voice pressure[8–11,14,15,18–21]. Thus, they cannot be used to correlate variations in vocal pattern and voice volume levels with human intention and mental states.

In this study, we suggest a novel methodology for the flexible and skin-attachable sensor to satisfy the essential requirements. First, we examined the neck skin vibrations that arise in human speech and ascertained that there is a linear relationship between the skin acceleration and the voice pressure. This finding suggests that the skin acceleration is an appropriate sensing parameter to measure human voice quantitatively. Then, we fabricated an ultrathin and conformable electronic skin by introducing polymers with low damping properties to sophisticated capacitive diaphragm structures that were designed by using a theoretical diaphragm model and the finite element method (FEM). The device exhibits a flat frequency response over the frequency range of 80~3400 Hz, which includes the standard telephony bandwidth telephony (300~3400 Hz) and the fundamental voice frequency range (80~255 Hz)[24]. This feature is attributed to the reduced damping that arises from the use of a fully crosslinked polymer material and the hole-patterned structure of the diaphragm. The device also exhibits superior vibrational sensitivity of $270\,mV\,g^{-1}$ for the range of neck skin vibrations, and a superior signal to noise ratio of more than 10 dB at $0.02\,g$ acceleration. This high sensitivity results from the low stiffness ($37.71\,N\,m^{-1}$) of the polymer diaphragm and its optimized structure, which enhance the degree of capacitance modulation, and from the interface circuit that effectively converts capacitances to voltages ($\sim 9\,V\,pF^{-1}$). The ultrathin structure provides precise detection by eliminating the vibrational distortions that can arise on rough and curved skin surfaces. Thus, our device can transduce human voice into electrical voltages with a flat frequency response and a high/linear sensitivity of $5.5\,V\,Pa^{-1}$. The device is the first flexible and skin-attachable sensor that can perceive human voices quantitatively by detecting the neck skin vibration. In addition, our device detects the human voice clearly in the presence of ambient noise or a mouth mask. The device exhibits not only high sensing performance compared to commercial microphones, but also skin-conformity and noise canceling functionality even in poor acoustic environments. With these merits, we successfully demonstrated the use of this device in several voice-recognition applications, namely voice authentication, voice remote control systems, and the healthcare monitoring of vocal cords.

## Results

**Analysis on neck skin vibrations for voice measurements**. Skin-attachable sensors recognize the human voice by monitoring neck skin vibrations. Such vibrations can be quantified with various parameters such as displacement, velocity, and acceleration. We first analyzed neck skin movements in order to identify a skin-vibration parameter that is highly correlated with voice pressure. We measured voice pressures and neck skin vibrations simultaneously while a person spoke at different volume levels with fundamental voice frequencies, which are the largest among numerous harmonics of human voice and represent the voice[25]. We chose the three frequencies of 100, 150, 200 Hz in the fundamental voice frequency range (80 to 255 Hz) with even intervals. Voice pressure and neck skin vibration were simultaneously measured while a person spoke (Supplementary Fig. 1a). The voice pressure was measured using a reference microphone placed 1 m in front of the mouth (Brüel & Kjaer, type 4192), and the measured time-domain voltage signal was converted into a frequency domain by a signal analyzer (Stanford Research Systems, SR785). Neck-skin vibration was measured accurately using a laser Doppler vibrometer (LDV; Polytec, OFV-5000). The velocity of neck vibration was measured as a voltage in the time-domain signal, and then converted to a frequency spectrum by a signal analyzer. The amplitude of the fundamental frequency was converted to three vibration parameters: displacement, velocity, and acceleration. Finally, we analyzed the correlation between voice pressure and corresponding skin vibration (Supplementary Fig. 1c–f). Our results show that there is a linear relationship between the voice pressure and the acceleration of neck skin with a proportionality constant of $22.2\,g\,Pa^{-1}$ and a high coefficient of determination of 0.983 (Fig. 1a and Supplementary Fig. 1e). This correlation was also found to be maintained at the fundamental voice frequencies and various sound pressure levels (40~70 $dB_{SPL}$ at 1 m[26,27]) for voices from whispering to shouting. Thus, the acceleration of neck skin vibrations is an appropriate parameter for characterizing the human voice. Our device is designed to transduce the accelerations of neck skin vibrations into electrical signals.

**Configuration of vibration-responsive electronic skin**. Our vibration-responsive electronic skin, conformally attached to the skin of neck, is shown schematically in Fig. 1b (the details of its fabrication are in the 'Methods" section and Supplementary Fig. 2). The output of the device in response to vibrational accelerations is based on changes in the capacitance between the electrodes on the diaphragm and the substrate. Our device has two electrodes placed in parallel with a gap of μm, which are distinct from two electrodes aligned on the same plane of capacitance proximity sensors that generally detect the surrounding objects. The device consists of a suspended diaphragm array: the diaphragms are electrically connected in parallel to accumulate the electric charges from the capacitors and thus to increase the output signal. As shown in Fig. 1c, the device is thin enough (<5 μm) to be conformally attached to a curved surface with a bending radius of 2.5 mm. Eight holes are patterned around the rim of each diaphragm to reduce the stiffness of diaphragm and

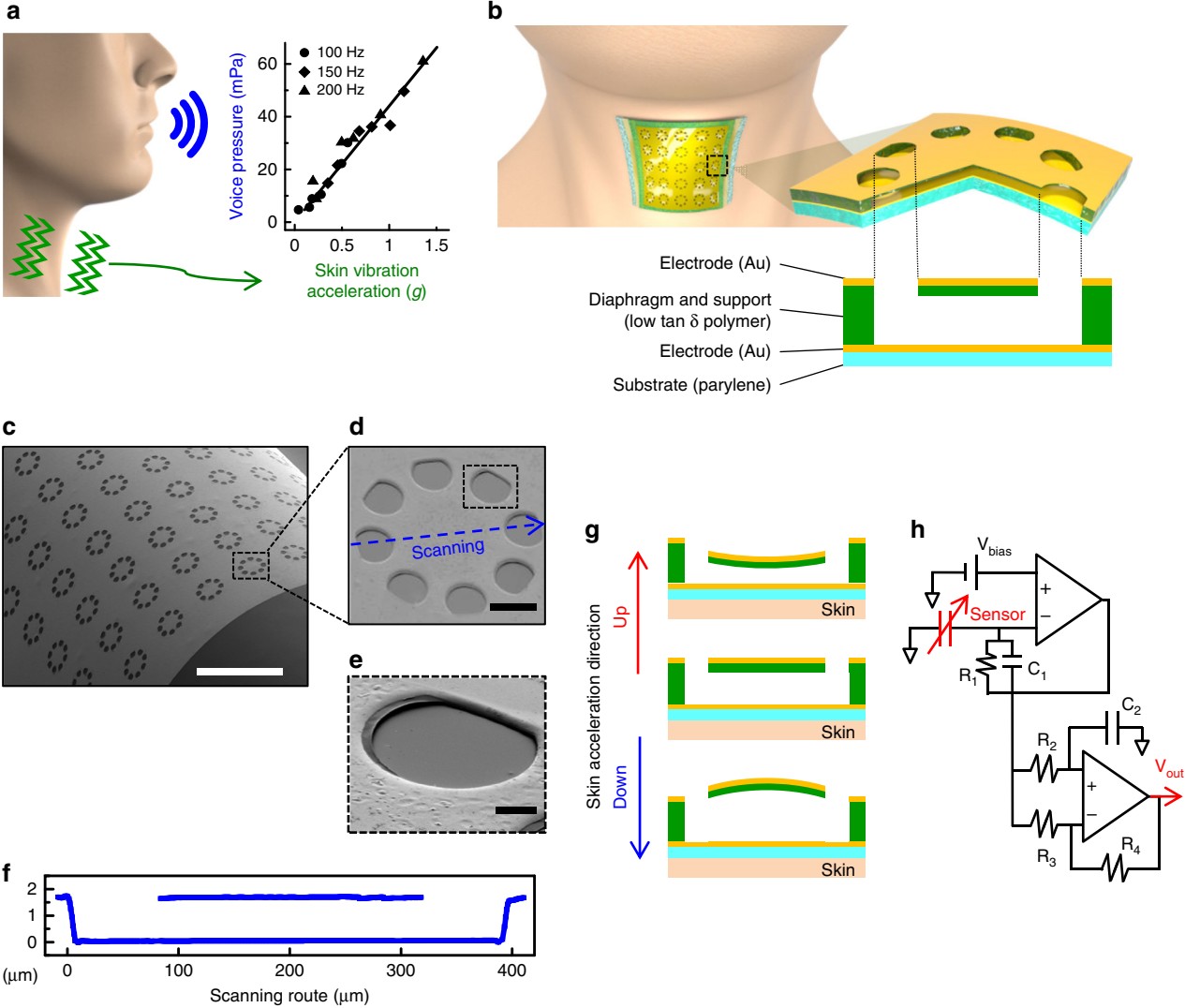

**Fig. 1** Schematic design of vibration-responsive electronic skin. **a** Measurement data of neck skin acceleration and voice pressure at three fundamental voice frequencies of 100, 150, and 200 Hz. **b** Illustration of our electronic skin attached on neck and the diaphragm structure. Human bust images in (**a**, **b**) were created based on a base-mesh, adapted with permission from https://www.deviantart.com/chrrambow/art/Bust-Basemesh-128960722. Copyright 2009 by Christian Rambow. **c** SEM image of diaphragm array on a curved surface with bending radius of 2.5 mm. **d** SEM images of hole patterns on the suspended diaphragm. **e** Magnified SEM image of the hole pattern shown in the dashed line rectangle of the SEM image (**d**). **f** Height profile of cross-sectional view for a suspended diaphragm structure. Scanning route is shown as a blue dotted arrow in the SEM image (**d**). **g** Schematic illustration during the movement of a diaphragm when the skin vibrates. **h** Interface conditioning circuit diagram connected to the sensor. Scale bars: 1 mm (**c**); 100 μm (**d**); 20 μm (**e**)

the air damping beneath the diaphragm (Fig. 1d). Each ultrathin diaphragm (<1 μm) is suspended on a 1.2 μm thick epoxy resin support (Fig. 1e), and its initial downward deflection is less than 100 nm (Fig. 1f and Supplementary Fig. 3). The aspect ratio of the air film under the diaphragm is approximately up to 400:1, which results in a large capacitance change when the diaphragm is deflected up/downward.

When the device is attached to the neck skin over the cricoid cartilage, it vibrates if a voice is generated in the larynx and articulator, as shown in Fig. 1g. Each diaphragm in the device moves up and down dynamically by the force of inertia resisting the vibration. The capacitance of the diaphragm is modulated by changes in the distance between the top and bottom electrodes. The capacitance of one diaphragm unit (lateral dimension: 400 μm) is on the order of a few pF, and its subtle variation is difficult to monitor without bulky measuring equipment. Therefore, a fixed voltage bias ($V_0$) is used to convert the capacitance

variation ($\Delta C$) to an electric charge flow ($\Delta Q = \Delta C \times V_0$). Since the diaphragms are connected in parallel, the charge is accumulated and converted to an output voltage signal by using an interface circuit with an amplifier (Fig. 1h and Supplementary Fig. 4). By considering the overall operational mechanism, we established a transfer function that relates the output voltage signal to the skin acceleration (the details are in Supplementary Note 1).

**Flat frequency response.** A flat frequency response, defined as uniform sensitivity over the entire input frequency range, is essential to the accurate sensing of voice pressure patterns. The frequency response of the device is affected principally by the mechanical resonance frequency and the material/structure damping, according to the lumped element model for a diaphragm structure[28] (Supplementary Note 1). The resonance frequency of the device needs to be more than several times higher

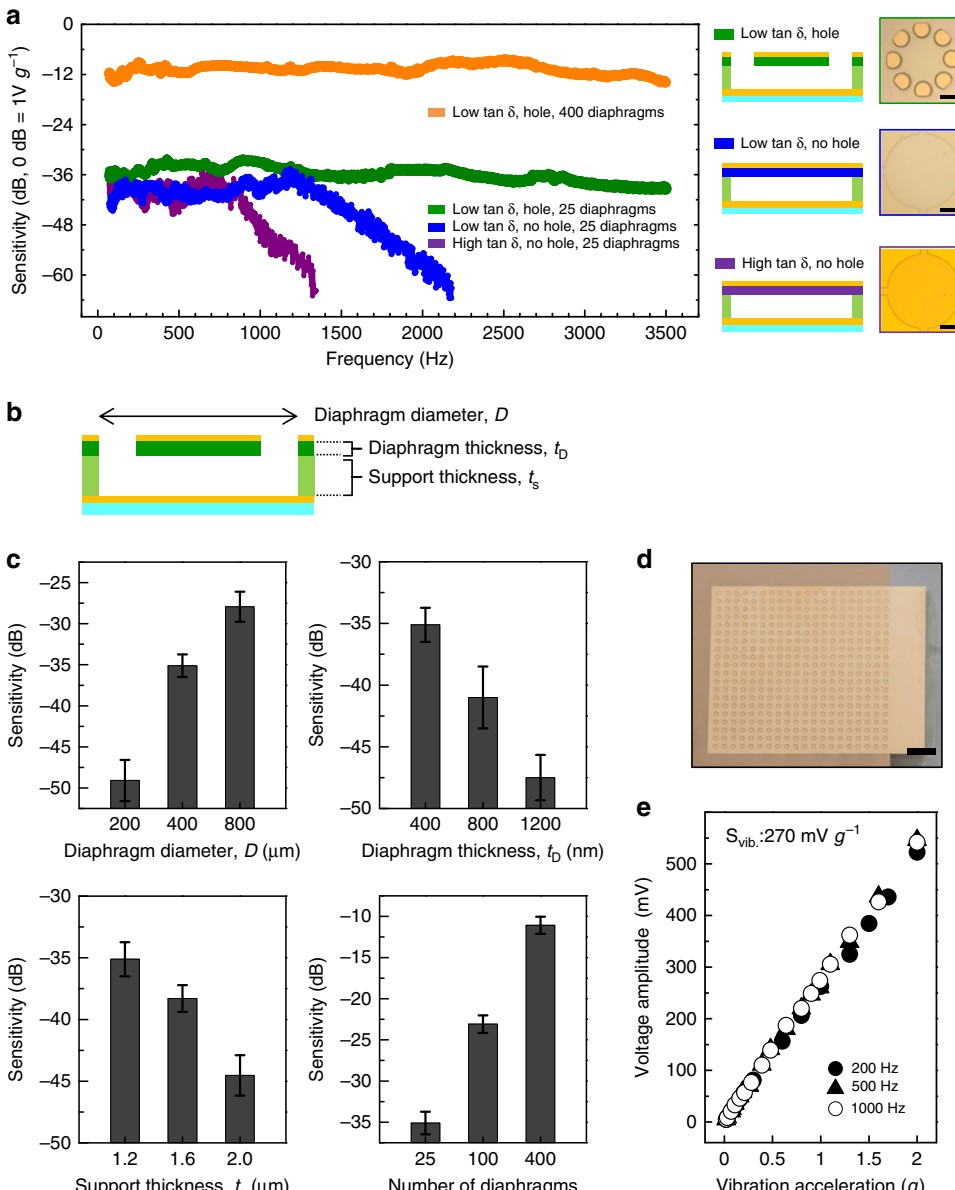

**Fig. 2** Effects of material and structure on the device performance. **a** Frequency response data for the devices with different material, structure, and number of diaphragms. The zero decibel line corresponds to a response of 1 V $g^{-1}$. Optical microscope images (Scale bars: 100 μm) and cross-sectional schematic drawings of diaphragms are shown in the right. **b** The schematic shows the cross-sectional structure of a diaphragm with air holes and its structural parameters. **c** Effects of diaphragm diameter; diaphragm thickness; support thickness; and the number of diaphragms on the vibrational sensitivity. For each graph, low tan δ polymer was used for the diaphragm material, air holes were patterned on the diaphragm, and the structural parameters were fixed as follows unless specified: diaphragm diameter of 400 μm, diaphragm thickness of 400 nm, support thickness of 1.2 μm and an array of 25 diaphragms. The error bars represent the s.d. of the sensitivity values for the measured frequencies. **d**, **e** Microscope image (Scale bar: 2 mm) (**d**) and output data (**e**) of an optimized device: an array of 400 low tan δ polymer diaphragms with air holes on 2 cm² area

than the maximum input frequency, otherwise the sensitivity fluctuates significantly near the resonance frequency and rapidly decreases henceforth[10,11]. However, an increase in the resonance frequency requires a diaphragm with a higher stiffness or lower mass, which reduces the sensitivity of the vibration sensor[28]. Considering this trade-off, we optimized the resonance frequency by using a hole-patterned diaphragm structure with the appropriate combination of mass and stiffness values (Supplementary Note 2 and Supplementary Fig. 5). Finally, we determined that this device exhibits its first mechanical resonance around 90 kHz by using the theoretical equation for the natural frequency of the device structure and performing an FEM simulation (Supplementary Fig. 6).

The effects of the material/structure damping on the frequency response of the device are shown in Fig. 2a. The vibration sensitivity of the device ($S_{vib}$) is defined as the output voltage signal ($V_{out}$) divided by the vibrational acceleration of the base ($a_{base}$). The frequency response sensitivity was defined as the output voltage of the device relative to that of the reference accelerometer (PCB Piezotronics, 352C33), which has a constant sensitivity of 100 mV $g^{-1}$ from 10 Hz to 10 kHz (Supplementary Fig. 7). An input vibration was precisely generated by using an electromagnetic vibration speaker (Newadin Technology, VBT-001). To remove any undesirable electromagnetic coupling with the speaker, the device was kept in an aluminum shielding box. The diaphragm material should maintain low loss factors (tan δ)

over the frequency range 80~3400 Hz to reduce mechanical damping. We utilized a fully crosslinked epoxy resin (SU-8) as diaphragm material. The monomer of the epoxy resin is based on four bisphenol-A units, which exhibit a distinct beta transition at $\simeq -100$ °C[23] and a glass transition over 150 °C, thereby having low tan δ at room temperature. In addition, we further reduced the tan δ of the polymer by fully crosslinking the epoxy binding sites during a hard baking process at a temperature of 240 °C. The fully crosslinked network structure prevents the oscillation and conformation flip of the phenyl rings in bisphenol-A[29,30]. Therefore, our fully crosslinked SU-8 exhibits a wider bandwidth of flat frequency response (Fig. 2a, blue) than poly(methyl methacrylate) (PMMA) with a similar mass density and stiffness but higher tan δ due to a rotating motion of the ester groups[23] (Fig. 2a, purple). According to a theory of squeezed-film air damping[31], an air film under a diaphragm, isolated from ambient air, has unfavorable effects on the diaphragm frequency response. As the volume of air film is decreased and the ratio of lateral dimension to height in the air film is increased over twenty times, the structural damping becomes more significant. For this reason, the polymer diaphragm without holes has a flat frequency response up to only 1300 Hz in spite of the reduced material damping (Fig. 2a, blue). To decrease the structural damping effects, we patterned eight holes in the diaphragm to ventilate an air beneath the diaphragm to the ambient air, which results in improved flat frequency response up to 3500 Hz (Fig. 2a, olive).

**Device design and vibration sensitivity**. The polymer material and hole-patterned structure contribute not only to a flat frequency response, but also to a sufficiently high $S_{vib}$ for voice pressures to be measured accurately because it reduces the stiffness of our device. We utilized a polymer in the diaphragm structure that has a lower stiffness than the inorganic materials. In the load-deflection model of a circular diaphragm[32], the stiffness of a diaphragm with an aspect ratio of several hundreds is affected predominantly by the residual stress of the diaphragm material (Supplementary Note 3). In our device, the diaphragm has a low stiffness of $44.71$ N m$^{-1}$ (Supplementary Fig. 5), because its residual stress is only 8.93 MPa (Supplementary Note 3), which is one or two orders of magnitude lower than those of the inorganic diaphragms conventionally used in MEMS[33,34]. In addition, the holes in the diaphragm reduce the stiffness to 84% with respect to the diaphragm without holes (Supplementary Fig. 5), by enhancing the movement of the rim of diaphragms. As a result, the holes increase $S_{vib}$ by 3 dB as shown in the frequency band at <900 Hz (Fig. 2a, olive and blue). We studied the effects on $S_{vib}$ of the structural parameters of the device: the diaphragm diameter ($D$), the diaphragm thickness ($t_D$), the support thickness ($t_S$), and the number of diaphragms in the array (Fig. 2b). As shown in Fig. 2c, $S_{vib}$ increases with increase in $D$, and decreases with increase in $t_D$ and $t_S$. The rate of increment in $S_{vib}$ decreases as $D$ increases, and the rate of decrement increases as $t_S$ increases. $S_{vib}$ is proportional to the number of diaphragms. In addition, considering the maximum number of diaphragms that can be fabricated in the same area, the array of diaphragms with $D = 400$ μm has a slightly higher $S_{vib}$ than the arrays with $D = 800$ μm and $D = 200$ μm (Supplementary Fig. 8). $S_{vib}$ is affected by the following factors: increases in the stiffness of the diaphragm and in the area of the air film under the diaphragm decreases $S_{vib}$; increases in the mass of the diaphragm and in the change in the capacitance with diaphragm deflection ($\Delta C/\Delta d_{cap}$) increases $S_{vib}$ (Supplementary Note 1). These experimental results can be explained in terms of the effects of varying each structural parameter on the $S_{vib}$ change factors (the details of this analysis are in Supplementary Note 4). In our

theoretical analysis, we measured the stiffness and $\Delta C/\Delta d_{cap}$ by applying a DC voltage between the upper and lower electrodes of the diaphragm structure and simulated the same conditions with FEM (Supplementary Figs. 9–11). The optimal device structure with the highest performance consists of an array of 400 diaphragms, each of which has the following parameters: $D = 400$ μm, $t_D = 400$ nm and $t_S = 1.2$ μm on 2 cm$^2$. This lateral dimension is large enough to cover the neck skin near the cricoid cartilage (Fig. 2d). The device has a high $S_{vib}$ of 270 mV g$^{-1}$ (Fig. 2a, orange), which is attributed to the hole-patterning and low stiffness of the polymer diaphragm and to the large and ultrathin diaphragms for enhanced capacitance modulation. The interface circuit connected to the diaphragm array also affects $S_{vib}$ when it converts the diaphragm capacitance into voltage; the conversion efficiency is approximately 9 V pF$^{-1}$ (Supplementary Fig. 4). Our device has approximately 10 times higher spectral noise level than that of the commercial accelerometer (Supplementary Fig. 12) due to the difference of electromagnetic shielding and electric wiring design. Nonetheless, our device exhibits a high signal-to-noise ratio of more than 10 dB for a vibration of 0.02 $g$ (Supplementary Fig. 13), which is the smallest neck-skin vibration arising in human speech. As shown in Fig. 2e, the device has a uniform $S_{vib}$ for the input vibration range 0.02~2 $g$, which corresponds to the range of neck skin vibrations of human speech (see the range of measured skin accelerations in Fig. 1a). When the device is on a vibrating base, $a_{base}$ can be linearly converted to the diaphragm deflection ($\Delta d_{cap}$), because the diaphragm has a constant stiffness (Supplementary Note 3). The deflection shape of the diaphragm is parabolic[35], so $\Delta d_{cap}$ and the corresponding $\Delta C$ value have a linear relationship (Supplementary Fig. 14). In addition, $\Delta C$ is correlated to $V_{out}$ with a constant gain, defined by the performance of the interface circuit (Supplementary Fig. 4). In summary, $a_{base}$ is linearly converted to $\Delta d_{cap}$, $\Delta C$, and finally $V_{out}$ in succession.

**Recording of sound vibrations**. The performance of our device was assessed by recording music from the electromagnetic vibration speaker (the details of the experimental set-up are in Supplementary Fig. 7). We composed a sheet music with two or three tones of adjacent frequencies corresponding to human voice and a sixteenth note in 80 beats per minute, as shown in Fig. 3a. The output of the device was recorded as voltage signals and transformed into frequency spectra (Fig. 3b; Supplementary Fig. 15a for broader frequency range). The same music, played under identical conditions, was also recorded with the reference accelerometer, which exhibits a uniform sensitivity of 100 mV g$^{-1}$ over the range 10~10,000 Hz (Fig. 3c; Supplementary Fig. 15b for broader frequency range). Our sensor device exhibited three times higher sensitivity than the commercial accelerometer. In addition, our device was found to have a flat frequency response for voice frequency range, because the frequency spectra from our device and the reference accelerometer show almost identical peak shapes. The performed sheet music is clearly distinguished based on the peaks in the frequency spectrum (Fig. 3b), because the device can detect complicated frequency spectrum as well. Therefore, we found that the vibration music was recorded successfully by our sensor device.

**Voice recognition on human neck skin**. We then analyzed the voice-recognition performance of the device, when it is attached to the neck skin (Fig. 4a). The device was attached near the cricoid cartilage, on which the magnitude of neck skin vibration was found to be the largest (the details of this analysis are in Supplementary Note 5 and Supplementary Fig. 16). We recorded the output voltage amplitude during human speech at volumes varying from 40 to

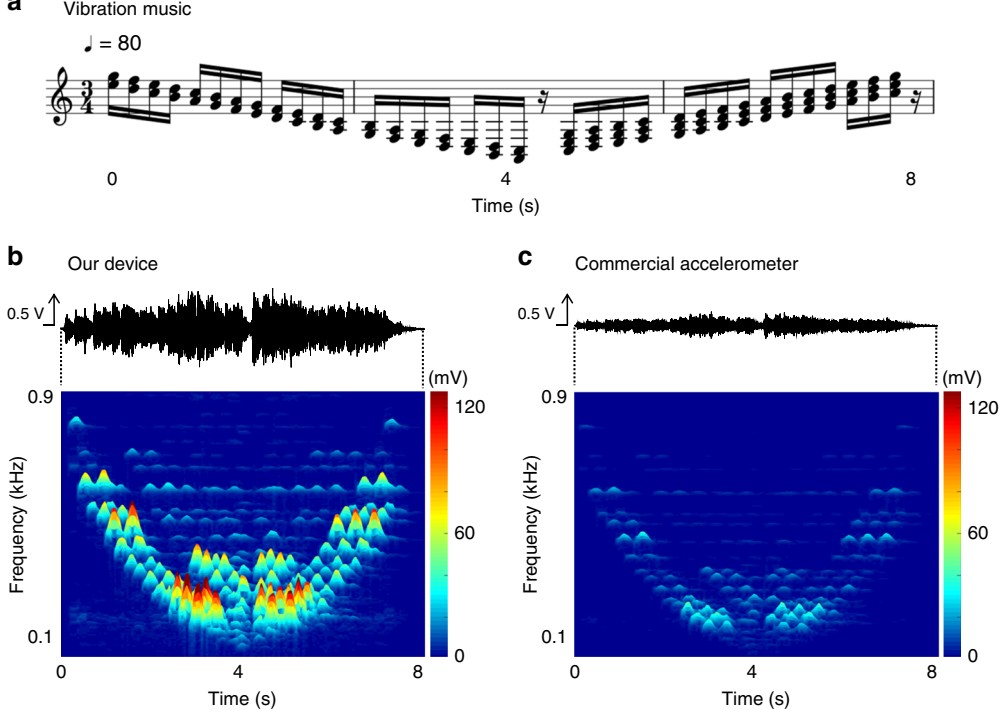

**Fig. 3** Comparative vibration recognition test with commercial accelerometer. **a** While a sheet music was played by a vibration speaker (Newadin Technology, VBT-001), vibrational output was detected from our device and commercial accelerometer. **b**, **c**, Output waveform and its frequency spectrum from our device (**b**) and a commercial accelerometer (sensitivity: 100 mV $g^{-1}$ for 10~10,000 Hz) (**c**) as a function of time

70 dB$_{SPL}$ and three representative fundamental voice frequencies: 100, 150, and 200 Hz. As shown in Fig. 4a, the device exhibits a high and linear sensitivity of 5.5 V Pa$^{-1}$ at these voice frequencies and all voice pressures: this performance represents an outstanding improvement over that of the flexible vibration sensors reported so far, which cannot recognize variations in vocal patterns and loudness due to their non-flat frequency response and insufficient sensitivity. Furthermore, our device exhibits voice-recognition ability that is comparable to that of commercial microphone systems including throat microphones[36]. The performance of our device is maintained on neck skin surfaces due to its skin conformity. Neck skin has a curvature of approximately 16 m$^{-1}$ and a roughness of tens of microns[37,38], so secondary spring system forms inevitably at the sensor-skin interface when a vibration sensor is attached on the skin, which results in the distortion of output values. However, our flexible device is based on a polymer with low stiffness and has an ultrathin structure (<5 μm), so can be conformally attached to neck skin (Fig. 4a, c), and can reduce a secondary spring effect. To quantitatively assess the benefits of this skin conformity, we compared the skin acceleration values obtained from our device and the commercial accelerometer during human speech, as shown in Fig. 4b. Reference data for actual skin vibrations were obtained by using contactless LDV without any secondary spring effects. The data obtained with our device and the accelerometer feature lower proportional constants than the data obtained with LDV, which is due to the sensitivity degradation that arises from the secondary spring system. The sensitivity of the accelerometer was degraded to 40%, because the skin is compressed when the accelerometer with the rigid and bulky structure should be fastened by strong adhesive tape (Fig. 4c). However, our device on neck skin still maintains more than 90% of its vibration sensitivity. In addition, our device is capable of detecting human voice clearly even under dynamic situations such as shaking the head back and forth (Supplementary Fig. 17). With this demonstration, we have confirmed that our device can detect neck skin vibrations

corresponding to human voices with diverse magnitudes and frequencies.

**Application: voice authentication and speech recognition**. Biometric systems can provide promising security methods because of the low risk of loss, sharing, and copying[39]. Voice recognition is an advanced biometric system: each entity has a unique voice pattern, and there are an infinite number of vocal passwords[39]. Our device with outstanding voice-recognition performance and skin conformity has potential uses in voice-controlled security systems. We demonstrated a prototype for voice security authentication created by using our device. After the device was connected to a speech-recognition module, the voice pattern "Siyoung log-in" was set as the login password (the details of this method are in the 'Methods" section). The voice authentication system was found to work successfully (Supplementary Movie 1): our device distinguished the unique voice sound waveform and frequency spectrum of each person's voice, even when other speakers uttered the correct voice password, as shown in Fig. 5b. More importantly, it was able to recognize the user's voice clearly even when a mouth mask was worn because its sensing mechanism is based on skin vibration (Fig. 5a, Supplementary Fig. 18 and Supplementary Movie 1); this capability reduces the risk of exposing the vocal password to others. Recently, speech recognition systems have been used widely in the control of computers and peripheral devices with voice commands. However, the utility of these devices, most of which receive vocal information encoded in air pressure waves, is limited in acoustically noisy environments as illustrated in Fig. 5a. Our device overcomes this problem by exploiting skin vibrations to recognize human voices. For example, the device was not influenced by the presence of ~62 dB$_{SPL}$ of noise, i.e. it maintained the same voice sound waveform and its frequency spectrum, whereas the reference microphone could not filter out the noise and exhibited recognition results that are different to those

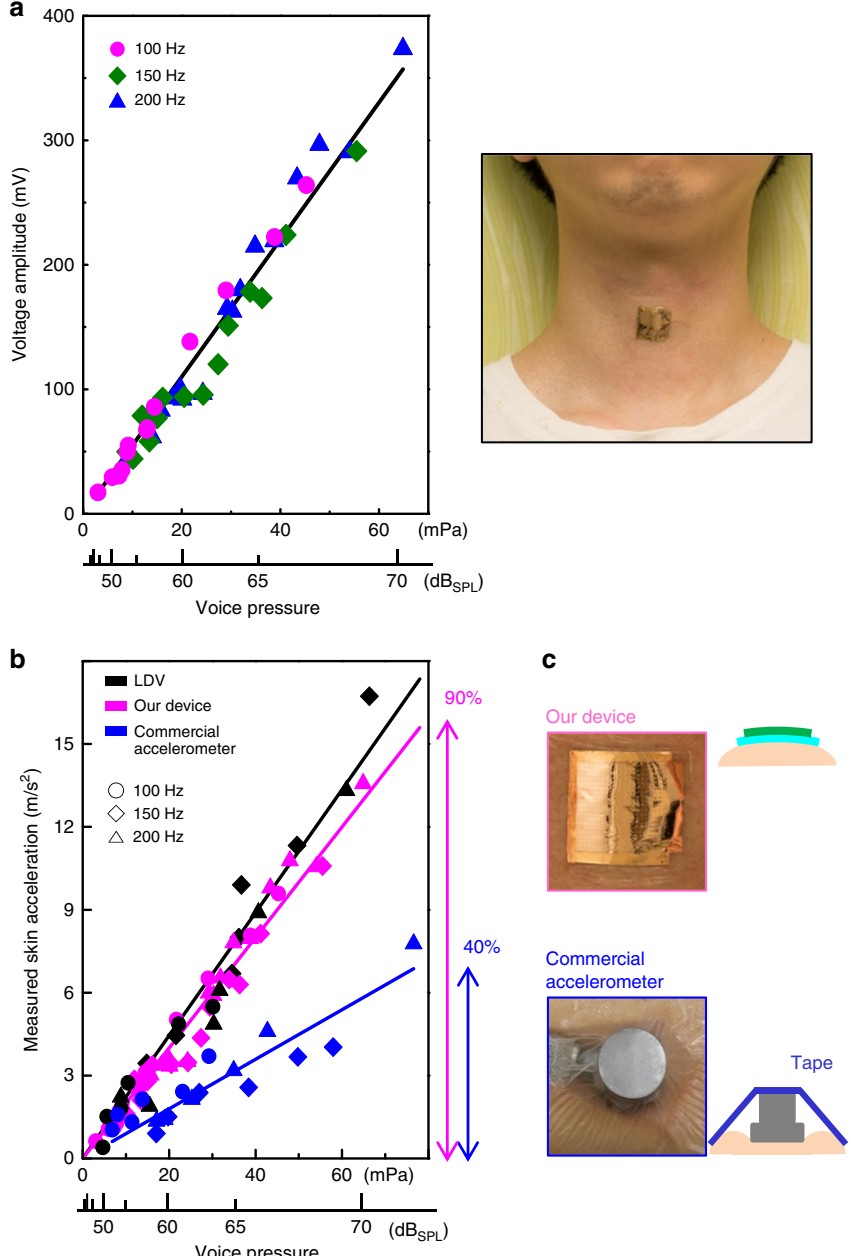

**Fig. 4** Voice-recognition performance on human neck skin. **a** Output voltage amplitude of our device as a function of voice pressure at different frequencies, and photograph of our device attached on the neck skin. **b** Comparison of measured the skin acceleration between laser Doppler vibrometer (LDV), our device and a commercial accelerometer, when a person spoke at various voice pressure and frequencies. The voice pressure is expressed in pascal (Pa) and decibel of sound pressure level (dB$_{SPL}$; 1 Pa = 94 dB$_{SPL}$). The percentage values on the right represent the reduced sensitivity due to rough and curved surface of neck skin. **c**, Photographs and cross-sectional schematic images of our device and the commercial accelerometer attached on the skin

obtained in a silent environment (Fig. 5c). As shown in Fig. 5a, we connected our device to the speech-recognition module and a remotely controlled lamp. The lamp was successfully controlled even in a noisy environment with audible music as well as in a quiet environment.

**Application: voice dosimetry.** More than one-third of the working population uses the voice as a primary tool at work[40]. Most of them worry about vocal conditions due to excess use of vocal cords and external environmental factors such as noise and the need to be heard over long distances. However,

no standard or convenient method is currently available for the diagnosis of vocal conditions in real time, although vocal health disorders have chronic characteristics and need sustained management. As shown in Fig. 6a, voice dosimetry is one approach to the monitoring of vocal health that operates by measuring vocal fold vibrations, vocal loadings, and the mechanical stresses inflicted on the vocal tract[41]; these results are obtained by analyzing the vocal patterns of human speech such as phonation time, voice frequency, and voice pressure. Our device can be used to quantitatively measure such vocal patterns with high and constant accuracy, even in the presence of ambient noise. Thus, it can provide a more reliable and

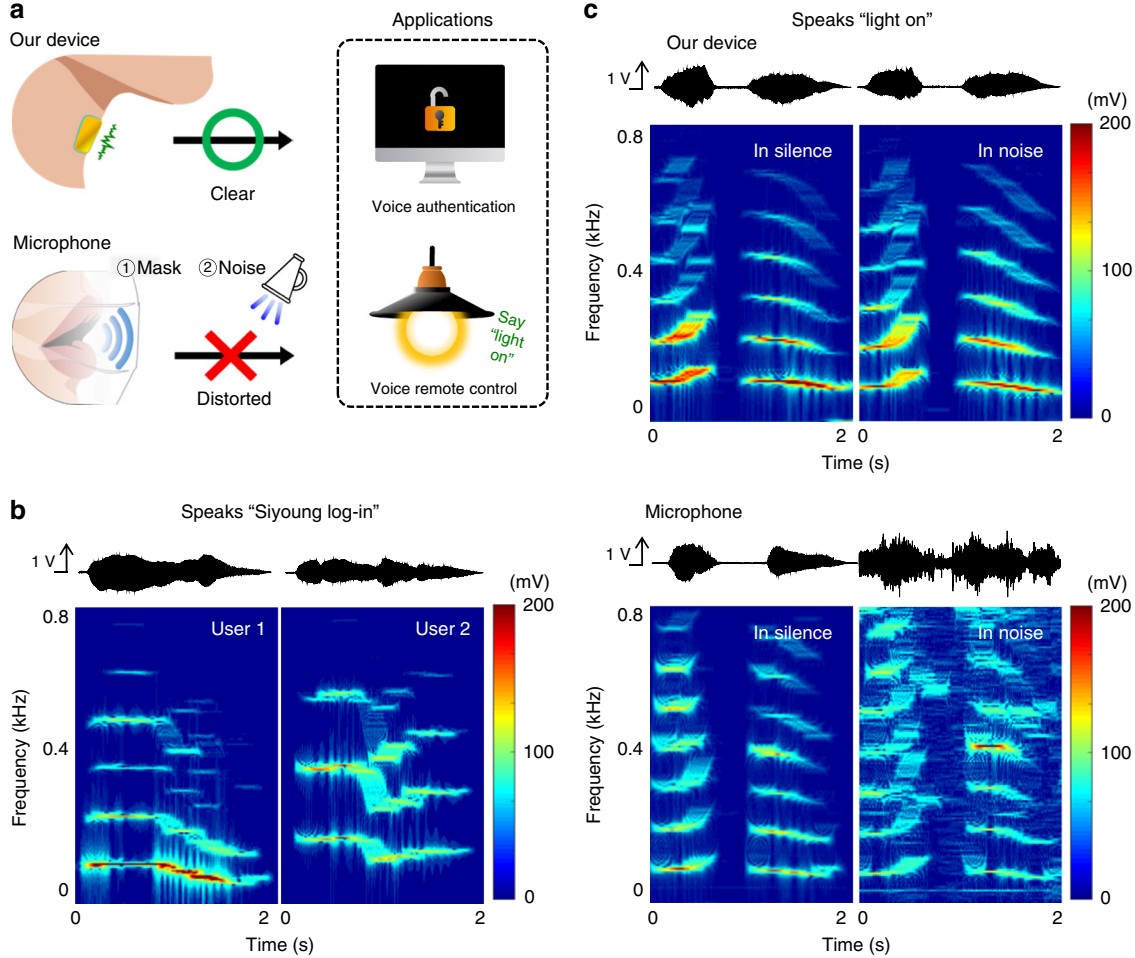

**Fig. 5** Voice authentication and speech recognition. **a** Schematic image of the comparison between our device and a reference microphone (Bruel & Kjaer, microphone type 4192, sensitivity of 1 V Pa$^{-1}$) for voice authentication and voice-controlled applications. **b** Output waveform and frequency spectrum measured by our device attached on neck when two users phonated "Siyoung log-in", respectively. Unique voice sound waveform and frequency spectrum of each user's voice are distinguishable. **c** Comparisons of waveform and frequency spectrum measured by our device and the reference microphone in silent and in noisy (62 dB$_{SPL}$) environments, when a person phonated "light on"

convenient tool for voice dosimetry than other voice-recognition devices. As shown in Fig. 6b–f, we obtained 4-minutes of vocal data from a male participant by using our sensor device and measured the acoustic parameters of his voice by using an established vocal theory[42] (the details of this method are in Supplementary Note 6). The participant vocalized loudly and frequently. The phonation time was measured by dividing the vocal data into 100-ms intervals and distinguishing between speaking and non-speaking periods (Fig. 6b, see Supplementary Equations (9) and (10) in Supplementary Note 6). The distributions of the fundamental frequencies and sound pressure levels (SPLs), which were obtained from the divided vocal data, were analyzed statistically (Fig. 6c, d). Occurrences of particular combinations of fundamental frequencies and SPLs were also plotted to show the patterns of vocal behavior more clearly (Fig. 6e). These results suggest that the male participant spoke incessantly at an average male voice frequency and at an SPL several times higher than the loudness of normal conversation (40~60 dB$_{SPL}$[26,27]). We determined the total distance of the vocal fold vibrations, which is the most important criterion of vocal health. We analyzed the amplitudes of the vocal fold vibrations from the divided vocal data, based on the vocal fold length, the measured SPLs and the fundamental voice frequencies (Fig. 6f, see

Supplementary Equation (12) in Supplementary Note 6). Then, the total distance was obtained by multiplying the amplitude during the phonation time by the fundamental frequency. The distance dose for speaking by the male participant was approximately 120 m (Table 1), which is less than 25% of the medical safety limit for protecting vocal fold tissue, i.e., 520 m[42]. This result implies that if the participant had spoken with the same voice properties for 15 more minutes, the vocal folds could have been damaged. We calculated the heat dissipated due to the viscosity of the vocal folds, which depends on the vibrational amplitude and the vocal fold thickness (see Supplementary Equations (13) to (15) in Supplementary Note 6). The amount of dissipated heat for the participant was 185.7 mJ cm$^{-3}$ (Table 1), which increased the temperature of the tissues by 0.05 °C. Therefore, the dissipation of heat does not have large immediate effects on the vocal fold tissues, but could cause problems during extended speaking. We performed the same voice dosimetry process for a female participant who spoke occasionally at a low volume and found that the participant maintained her vocal health stably (Supplementary Fig. 19). We successfully performed diagnoses of the vocal health of the two participants by quickly and precisely tabulating the key voice dose parameters with our sensor device (Table 1).

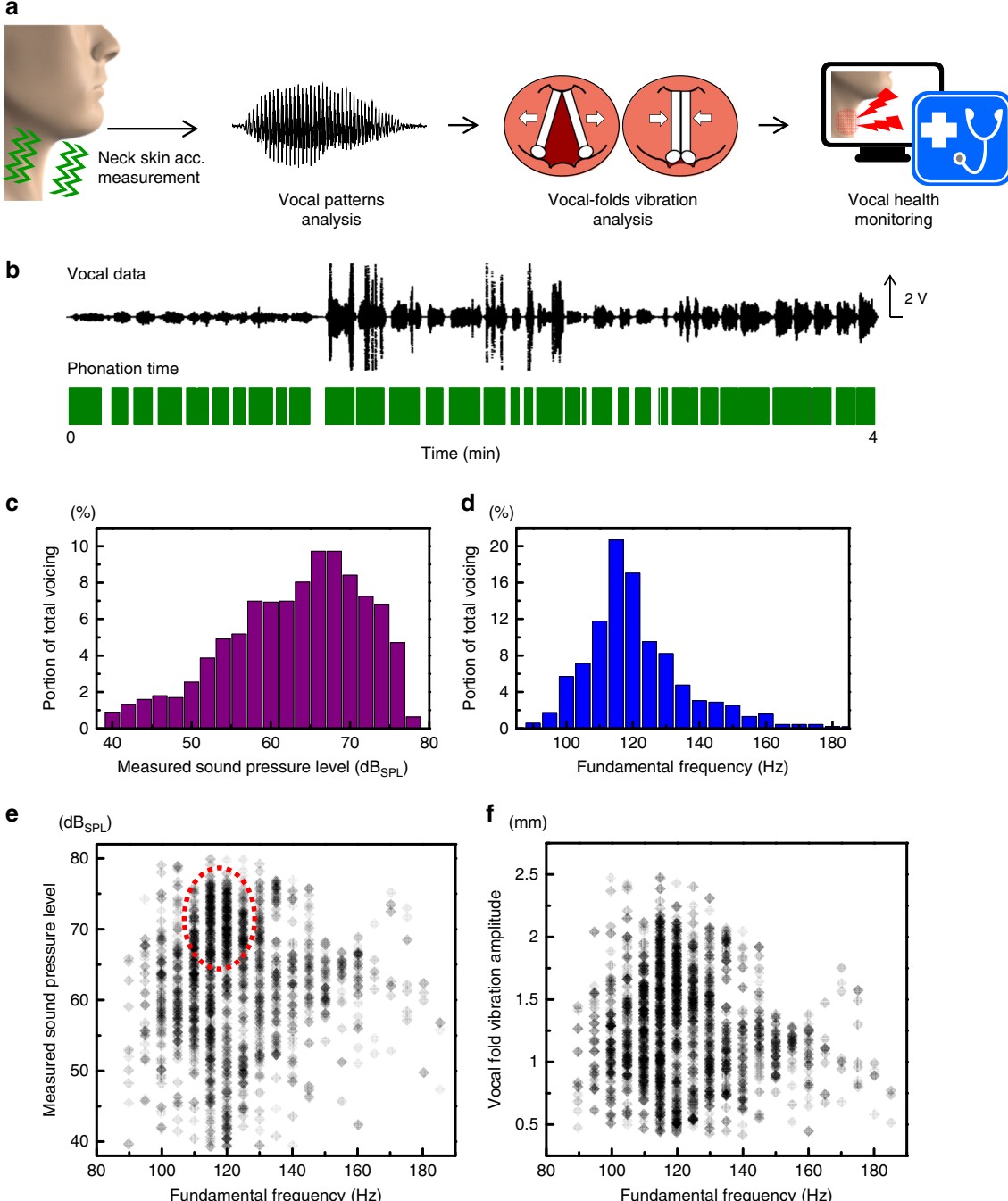

**Fig. 6** Voice dosimetry by detecting neck skin vibration. **a** Schematic overview of voice dosimetry by measuring neck skin acceleration and analyzing vocal patterns. Human bust images in (**a**) were created based on a base-mesh, adapted with permission from https://www.deviantart.com/chrrambow/art/Bust-Basemesh-128960722. Copyright 2009 by Christian Rambow. **b** Vocal waveform data measured by our device, and phonation time analyzed by dividing the vocal data into 100-ms intervals and distinguishing between speaking and non-speaking. **c**, **d** Histogram of sound pressure levels (**c**) and fundamental voice frequencies (**d**), each of which is extracted from the measured vocal data in (**b**). **e** The occurrence of particular combinations of fundamental frequency (horizontal axis) and sound pressure level (vertical axis) from the vocal data in (**b**). Dotted red circle represents the most frequent occurrence of this particular analysis example. **f** The amplitude of vocal fold vibrations as a function of fundamental voice frequency based on the vocal fold length and the measured sound pressure levels (see Supplementary Equation (12) in Supplementary Note 6)

## Discussion

We have developed an ultrathin and skin-conformable voice-recognition electronic skin that can quantitatively characterize the human voice. We have introduced a polymer with low stiffness and low damping properties into the large and ultrathin diaphragm structure. Thus, the device exhibits a high sensitivity and a flat frequency response over the range of voice frequencies, compared to commercial voice-recognition devices. The device recognizes the human voice clearly in the presence of ambient noise or a mouth mask, since the sensing mechanism is based on skin vibration rather than sound pressure. Furthermore, the device accurately detects voices throughout the frequency spectrum by detecting skin acceleration, which is highly and linearly correlated to voice pressure; this property ensures its high

### Table 1 Summary statistics of voice dosimetry parameters of two participants

| Measure | Participant 1. Male | Participant 2. Female |
|---|---|---|
| Performance Time (s) | 244 | 331 |
| Time dose (s) | 192.3 | 41.7 |
| Phonation Percent (%) | 78.8 | 12.6 |
| Average F0 (Hz) | 121.1 | 222 |
| Average SPL ($dB_{SPL}$) | 67.3 | 35.6 |
| Distance dose (m) | 119.6 | 9.8 |
| Energy dissipation dose ($mJ cm^{-3}$) | 185.7 | 1.3 |
| Temp. Increase in vocal fold (°C) | 0.05 | 0.0003 |

compatibility with current voice-recognition systems and human ears. The device exhibits a comfortable fit on skin because of its low stiffness and ultrathin structure. These advantages mean that our device exhibits significant potential as a next-generation voice-recognition device for HMI and IoT applications that require precise vocal information even in poor acoustic environments.

## Methods

**Device with hole-patterned diaphragms**. The overall fabrication process consists of the sequential deposition of a sacrificial layer, a top electrode, a diaphragm, and the diaphragm support onto a cleaned glass wafer, followed by the etching of the sacrificial layer. Then, the sample is separated from the glass wafer, flipped and transferred onto a polymeric substrate with the bottom electrodes (Supplementary Fig. 2). A Ti/Cu thin film (10/150 nm) was thermally deposited onto a cleaned glass wafer as a sacrificial layer. For the top electrode, a Ti/Au/Ti layer (3/35/7 nm) was thermally deposited and patterned with a lift-off process. The uppermost Ti layer (7 nm) acts as a promotor of the adhesion between Au and the polymeric diaphragm surface[43]. An epoxy resin diaphragm (360 nm) was formed by spin coating a diluted solution (Micro Chem., SU-8 2015) and by then patterning the resulting film into a circular array with air holes. The sample was then baked at 240 °C to fully crosslink the epoxy groups of SU-8 in order to achieve the desired damping properties. An additional SU-8 layer (1.2 μm) was spin coated onto the diaphragm and patterned for diaphragm support. The sacrificial Cu layer was then etched by using an organic/aqueous biphasic solution[44] of hexane and Cu etchant (Transene, CE-100) diluted with IPA (8 mol%). This method keeps the interface energy sufficiently low to preserve the structure of the thin film, when the film separated from the glass carrier wafer[45]. The separated layer was rinsed with a biphasic solution of hexane and deionized water diluted with IPA (8 mol%). The electrode/diaphragm/support layer was scooped using a polyarylate film with a square hole and dried in air. The polyarylate film (Ferrania technologies, PAR) was used due to its excellent mechanical and thermal stability and cutting formability for the lamination of the electrode/diaphragm/support layer. To fabricate the bottom electrode, Al (100 nm), parylene (3 μm), and Ti/Au (3/35 nm) were sequentially deposited in vacuum onto a glass wafer. Parylene was used as a polymer substrate due to excellent mechanical stability, bendability, thickness uniformity, and low damping property[46,47]. A thin film of SU-8 (20 nm) was spin coated onto the Au as an adhesive layer[48]. The dried electrode/diaphragm/support layer was then flipped upside down, placed onto the bottom electrode wafer, and pressed while heating at 120 °C. The layer including diaphragms suspended on the square hole of the polyarylate film was transferred onto the bottom electrode wafer[49]. The two samples were conformally adhered by cross-linking the epoxy groups on the SU-8 adhesive layer. After the sacrificial Al layer had been removed with Al etchant, the top and bottom electrodes were wired by performing silver pasting and soldering to connect them to an external interface circuit board for electrical characterization.

**Device with diaphragms without air holes**. An SU-8 diaphragm device without air holes was fabricated with the same process as described above, except that air holes were not prepared in the diaphragms. To admit air into the space under each diaphragm during the lamination process, narrow lines (width: 15 μm; depth: 1.2 μm) were additionally patterned onto the diaphragm support for ventilation (Fig. 2a, blue and purple OM images). To fabricate the device with a PMMA diaphragm, PMMA powder (Sigma Aldrich, Mw: 996k) was dissolved in chlorobenzene (50 g L$^{-1}$). Ti/Cu (10/150 nm) was thermally deposited onto a cleaned glass carrier wafer as a sacrificial layer. A PMMA film (360 nm) was prepared by spin coating the PMMA solution onto the glass carrier, which was then annealed in an oven at 120 °C for 1 h. Ti/Au/Ti (7/35/3 nm) layers were thermally deposited, then the Cu sacrificial layer was removed by using the Cu etchant. The PMMA/electrode sample was scooped with a polyarylate film with a square hole, then dried

in air. A diaphragm support layer with ventilation line patterns was placed on the bottom electrode, then the PMMA/electrode was laminated onto the bottom electrode with heating at 120 °C, which is higher than the glass transition temperature of PMMA (105 °C)[49].

**Characterization of neck skin vibrations**. The reference microphone has an uncertainty level of approximately ± 12.2 % (1 dB) at the tested frequency range. The laser Doppler vibrometer exhibits a resolution of 0.02 μm s$^{-1}$ with VD-09 decoder. The signal analyzer (sample acquisition time: >200 ms), converting time-domain voltage signal to the frequency domain, has an uncertainty of approximately ±2.3 % (0.2 dB) for amplitude measurement. A tiny piece of laser mirror sheet was softly attached onto the skin above the cricoid cartilage, which vibrates when a human speaks (Supplementary Fig. 1b). The exact position of the mirror was defined, based on a previous vocal study[50] for the mapping of neck surface vibrations during human speech. We adjusted the angle of the mirror to be perpendicular to the source of the laser Doppler vibrometer for accurate and stable focus of the laser source.

**Characterization of the device**. A cross-sectional profile and a 2D contour of the suspended diaphragm were obtained by using a 3D profiler (Veeco Touson, Wyko NT1100). The deflection at the diaphragm was measured with the same 3D profiler while applying a DC voltage between the top and bottom electrodes. Capacitance-voltage data were obtained by using an LCR meter (Keysight, E4980A). Simulations under the same conditions were performed by using COMSOL Multiphysics 5.2a. The real-time fast Fourier transform (FFT) of the output data was carried out by using a signal analyzer (SR785), while the vibration speaker was operated at a specific acceleration.

**Attachment of the device on the neck skin**. To attach the sensor device completely onto the neck skin, we applied a bio-compatible adhesive (LP-001, ABLE C&C Co., Ltd). In addition, an ultrathin and transparent medical tape (Tegaderm$^{TM}$ Film 1622W, 3M) was attached over the edge of the sensor and the metal lines connected to the interface circuit. Only a minimal amount of adhesive (~10 mg) was used to attach the device stably so that the adhesive does not affect the sensor operation. The bio-compatible adhesive is one of the cosmetics ingredients and can be easily removed by a make-up remover. The medical tape is widely utilized for skin attachment. Therefore, we did not experience a skin irritation or itch after three hours of wearing (Supplementary Fig. 20).

**Demonstration of applications**. The output waveform of the device was measured as a voltage by using a semiconductor device parameter analyzer (Keysight, B1500A) and the frequency domain was analyzed with FFT by using Matlab (MathWorks, R2016a). The sampling frequency of the time-domain signal was 10 kHz, window length was 0.2 s (data length is 2000), overlap was 0.05 s, and no window function was not used during FFT. In the voice-recognition experiments, we used a voice-recognition module (VeeaR, EasyVR 3) and a microcontroller board (Arduino, Uno R3) (Supplementary Fig. 21). The "Wake up" command was set with a low-security level for every users to operate the speech recognition device at first. The "Siyoung log-in, open the door" command was set with a high-security level so that people can be identified. If user authentication is succeeded, the door is opened by the motor connected to the voice-recognition module and the green light is turned on. The speaker connected to the module delivers the necessary information to users.

## Data availability

All the data that support the findings of this study are available from the corresponding authors upon reasonable request.

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

## Acknowledgements

This work was supported by a grant (Code No. 2012M3A6A5055728) from the Center for Advanced Soft Electronics under the Global Frontier Research Program of The Ministry of Science and ICT, Korea.

## Author contributions

S. Lee, J. Kim, I. Yun, G. Y. Bae, I. Yi, Y. Chung, and K. Cho designed the experiments. S. Lee, J. Kim, I. Yun, G.Y. Bae, D. Kim, S. Park, W. Moon, Y. Chung, and K. Cho performed experiments and analysis. S. Lee, Y. Chung, and K. Cho wrote the paper.

## Additional information

**Competing interests:** The authors declare no competing interests.

