## [Peer Review File · Nature Communications]

Reviewers' comments:

Reviewer #1 (Remarks to the Author):

This manuscript demonstrates a skin attachable voice recognition sensor with a linear sensitivity and high accuracy over the range of voice frequencies. The device utilizes an ultrathin and low stiffness diaphragm with uniform hole patterns, which provides the flat frequency response and high sensitivity compared to other conventional acoustic sensors. This study also shows a systematic study on the structural effect on the acoustic sensing capability. Finally, the authors demonstrated that the acoustic sensor can be applied for the security authentication, remote control systems, and vocal healthcare. I recommend the publication of this manuscript in the journal of "Nature Communications" if the authors address the following minor issues.

1. For the acoustic sensors, the capacitive-type sensor was used in this manuscript. Authors demonstrated that the device was not affected by the acoustic noise. However, the capacitive-based sensor can be affected by the surrounding electric fields, similar to the capacitive proximity sensors. Authors need to add some explanations or experimental results to clarify this issue.
2. In the calculation of a stiffness of a diaphragm, how authors obtained Young's modulus of the diaphragm? The Young's modulus of a thin membrane can be very different from that of a bulk material.
3. During the fabrication of hole-patterned diaphragm, the electrode/diaphragm/support layer was scooped using a polyacrylate film. Is there any special reason for the use of a polyacrylate film? Is it because of adhesion? When the diaphragm is flipped over and attached on another electrode, the adhesion between the diaphragm and polyacrylate film should weak enough for the facile transfer of a diaphragm.
4. There is a layer of 3 micrometers thick parylene between the diaphragm and the neck skin. The vocal vibration should pass through this parylene layer. Why the authors utilize the parylene layer? Is there any effect of this parylene layer on the vibration damping and vocal sensing performance? In addition, how was the device attached on the neck skin? Is there any additional adhesion layer?
5. In Figure 3, can authors compare the frequency spectrum of the original sound and the spectrum detected by the device?
6. Authors demonstrated a skin-attachable voice recognition sensor. For the broader applications, it is recommended that authors provide the performance of their sensor for the detection of acoustic sounds over the broader frequency range (4-12 kHz) than the vocal frequency range (below 3400 Hz).

Reviewer #2 (Remarks to the Author):

The manuscript reports an interesting work with clear writing. However, the novelty of the work seems not particularly strong. It seems that work is just a finding about a good voice sensor and the main contribution of this work is in the use of fully-crosslinked thin polymer and hole-patterned structure as diaphragm. There is little new knowledge developed. I would suggest the authors clearly state the novelty and major contributions of this work.

Other comments

1. There are lots of research works on using variable capacitors to detect vibration and sound, as well as for energy generation. Please give a systemic summary about the previous arts in variable capacitor for vibration sensors and sound detection.
2. The basic preparation and test methods should be presented in the main text.
3. The authors attributed the excellent flat frequency response of the devices to role of the fully crosslinked polymer diaphragm with a hole patterned structure in reducing damping. However, they fail to explain why.
4. Page 7 the sentences "...Most previous flexible and skin-attachable sensors consisted of

viscoelastic... material damping limits sensitivity as the input frequency increases." are background, they should be moved to the introduction section.

Reviewer #3 (Remarks to the Author):

This work presents an ultrathin device for voice sensing and recognition via acceleration measurement on human cricoid cartilage. Applications including voice authentication, speech recognition and voice dosimetry for vocal health monitoring are demonstrated. Taking advantage of specially designed structures, the device is described to show superior performances comparing to the commercial accelerometer and microphones under several application scenarios. However, several significant details including the wearability of the device, vocal pressure-skin acceleration relation characterization, and the performance characterization are not described in a sufficiently clear manner. The manuscript would be stronger if the author can address the following questions and supplement that information to the current content:

1. The author mentioned that the human neck skin possesses a curvature of approximately 16 m⁻¹. However, the neck skin should also have a curvature distribution, and the curvature near cricoid cartilage should be naturally higher than other parts. Furthermore, varying with subjects, the local surface near cricoid cartilage may not be developable and could be dynamic during speaking and motion. Can the device guarantee the conformity under all these situations?

2. How is the device fixed on the human neck? Is there any adhesive material on the substrate to prevent the device from delamination? How long can a subject wear the device? Will there be skin irritation or itch after a certain period of wearing? The authors are suggested to add information on those aspects.

3. In supplementary note 1 and supplementary figure 1, the authors mentioned a laser Doppler instrument associated with a tiny laser mirror was used for neck skin vibration measurement. The voice was also recorded by a microphone simultaneously to correlate with the vibration measured.

(1) What is the relative size of the mirror comparing to the cricoid cartilage? If the mirror is smaller, will its position influence the result?

(2) In supplementary figure 1c, why are the three certain frequency values (100, 150 and 200 Hz) chosen for analysis?

(3) What are the uncertainty levels of both measurement methods?

4. In the manuscript, spectrums are frequently used for device performance comparison, and the author mentioned that the fast Fourier transform (FFT) was used for joint time-frequency analysis.

(1) What is the sampling frequency of the time-domain signal? What are the parameters, including window length, overlap, and window function used for FFT?

(2) The author mentioned the device exhibited a sufficient frequency resolution and short response time as described in supplementary figure 14, which is not clear to the reviewer. To the reviewer's knowledge, the parameters of FFT will significantly influence the temporal and frequency resolution results if they are not chosen properly. A further detailed and systematic study is required to characterize the temporal and frequency resolution of the device.

(3) There are two different plotting styles of the spectrums (figure 3b,c and supplementary figure 14 belong to one, figure 5 and supplementary figure 15 belong to the other one). The reviewer suggests that the author should unify these two plotting styles.

5. In figure 3 and supplementary figure 7, the author used a commercial speaker to characterize the device performance compared with a commercial accelerometer.

(1) In this experiment, the author horizontally laminated the device on the speaker. The orientation of the device is different when it is worn on the subject's neck. Will the gravity affect

the performance of the device under these two different situations?

(2) The author chose two periods of music, "Pachelbel's Canon in C" and "knocking on the heaven's door" for performance characterization. However, the total frequency range of these two audios showed in the manuscript only covers 130 ~ 800 Hz, which is still far from the whole audible frequency range. The reviewer suggests that the author should conduct experiments in broader frequency ranges.

(3) For the comparison to the commercial accelerometer, the author was more focused on the sensitivity differences. The reviewer suggests that the author should also compare the spectral noise level with that of the commercial accelerometer, which is also a significant characteristic.

6. In figure 2a, the author showed the sensitivity-frequency relation of devices with holes made by low loss factors. A measurement of the device with high loss factors and holes should also be added to facilitate the comparison.

RESPONSE TO THE REVIEWERS' COMMENTS

(Manuscript ID: NCOMMS-18-33828)

Reviewer #1

Overall Comment: This manuscript demonstrates a skin attachable voice recognition sensor with a linear sensitivity and high accuracy over the range of voice frequencies. The device utilizes an ultrathin and low stiffness diaphragm with uniform hole patterns, which provides the flat frequency response and high sensitivity compared to other conventional acoustic sensors. This study also shows a systematic study on the structural effect on the acoustic sensing capability. Finally, the authors demonstrated that the acoustic sensor can be applied for the security authentication, remote control systems, and vocal healthcare. I recommend the publication of this manuscript in the journal of “Nature Communications” if the authors address the following minor issues.

Response: We are grateful to the reviewer for taking time to review our manuscript and for positive assessment of our work.

Comment 1: For the acoustic sensors, the capacitive-type sensor was used in this manuscript. Authors demonstrated that the device was not affected by the acoustic noise. However, the capacitive-based sensor can be affected by the surrounding electric fields, similar to the capacitive proximity sensors. Authors need to add some explanations or experimental results to clarify this issue.

Response 1: We agree with the reviewer’s comment that our device could be affected by the surrounding electric fields. Capacitive proximity sensors detect target objects based on a change in the electric field profile, which is generated between two electrodes. These electrodes are generally made on the same plane (*Sensors for Industry Conference 2005*, 22-26) so that the electric field is easily influenced by the surrounding objects. Simulation results confirm that a large portion of the electric field exists in the free space far from the electrodes when they are aligned horizontally as shown in Figure R1a. However, our device has the structure with two electrodes placed in parallel with a gap of μm ; a strong electric field is generated between two electrodes (the right of Figure R1b), and relatively weak electric field is only generated around the edges of two electrodes as shown in the center of Figure R1b. Therefore, our device structure is distinct from a capacitance proximity sensor as the electric field remains almost identical when an object approaches to the electrodes. Based on the discussions above, we have added some explanation in the revised text.

- Revised main text, page 6, line 131-133

“Our device has two electrodes placed in parallel with a gap of μm , which are distinct from two electrodes aligned on the same plane of capacitance proximity sensors that generally

detect the surrounding objects.”

Figure R1. Comparison simulation for electric fields surrounding two electrodes horizontally aligned (a) and vertically aligned (b), respectively. The structure of (a) and (b) represents a general capacitive proximity sensor and our device, respectively. Both of (a) and (b) show schematic illustrations for geometry (left), simulation results for overall surrounding electric fields (center), and electric fields in the red dotted box as a magnified view (right). The thickness of biased and ground electrodes was set to be 100 nm. In the magnified views on the right, y-axis is exaggerated 10 times more than x-axis in scale so that the contour of electric field change can be easily distinguished.

Comment 2: In the calculation of a stiffness of a diaphragm, how authors obtained Young’s modulus of the diaphragm? The Young’s modulus of a thin membrane can be very different from that of a bulk material.

Response 2: We agree with the reviewer’s comment that Young’s modulus of thin membrane can be 2 ~ 3 times lower/larger compared to that of bulk material (*Diamond and related materials* **2001**, 10, 2069-2074 and *Polymer* **2016**, 87, 114-122). However, in our circular diaphragm structure, residual stress has more dominant effects on the stiffness value than Young’s modulus. In equation (4) of Supplementary Note 3, the first term with residual stress term is more than three orders of magnitude higher than the second and third terms with Young’s modulus, because the radius is much larger ($\geq 200 \mu\text{m}$) than the thickness ($\leq 500 \text{ nm}$). For this reason, we analyzed the diaphragm characteristics with residual stress, rather than Young’s modulus, based on the related theories and experiments in Supplementary Note 3.

Comment 3: During the fabrication of hole-patterned diaphragm, the electrode/diaphragm/support layer was scooped using a polyacrylate film. Is there any special reason for the use of a polyacrylate film? Is it because of adhesion? When the diaphragm is flipped over and attached on another electrode, the adhesion between the diaphragm and polyacrylate film should weak enough for the facile transfer of a diaphragm.

Response 3: We used polyarylate film, not polyacrylate film. For the lamination of the electrode/diaphragm/support layer, a carrier film with square hole and heating process up to 120 °C are required. Therefore, we used a polyarylate film (Ferrania technologies, PAR) due to its excellent mechanical and thermal stability and cutting formability. The thickness and Young's modulus of the film are 200 μm and 2 GPa. The glass transition temperature is over 300 °C. We have added the reason for the use of a polyarylate film briefly in the methods section.

- Revised main text, page 18, line 419-421

“The polyarylate film (Ferrania technologies, PAR) was used due to its excellent mechanical and thermal stability and cutting formability for the lamination of the electrode/diaphragm/support layer.”

The electrode/diaphragm/support layer was adhered to the carrier film by van der Waals forces. The center region of the layer including diaphragms was suspended on the square hole. When the diaphragm was flipped over and attached on the target surface, only the suspended center region was transferred, and the region of the layer was remained on the carrier film. Detail in the transfer method is provided in our previous work for water-free transferring of graphene (*Advanced Materials* **2014**, 26, 3213-3217). We have added the detail of transfer process briefly in the methods section.

- Revised main text, page 18, line 427-429

“The layer including diaphragms suspended on the square hole of the polyarylate film was transferred onto the bottom electrode wafer⁴⁹.”

- Added reference

(49) Kim, H. H. & Chung, Y. *et al.* Water-Free Transfer Method for CVD-Grown Graphene and Its Application to Flexible Air-Stable Graphene Transistors. *Adv. Mater.* **26**, 3213-3217 (2014).

Comment 4: There is a layer of 3 micrometers thick parylene between the diaphragm and the neck skin. The vocal vibration should pass through this parylene layer. Why the authors utilize the parylene layer? Is there any effect of this parylene layer on the vibration damping and vocal sensing performance? In addition, how was the device attached on the neck skin? Is there any additional adhesion layer?

Response 4: We used parylene as a polymer substrate to support the diaphragm structure and to adhere to the skin. Parylene is one of representative substrate materials for electronic skin due to its excellent mechanical stability, bendability and flexibility (*Advanced Materials* **2018**, 30, 1803388).

The uniform thickness of the substrate is an essential for the diaphragm structures to be stably transferred without wrinkles and folds. Because parylene can be thermally evaporated, its thickness is uniform than other polymer layer based on solution process, and no edge bead exists.

Parylene was chosen as a substrate material because of its excellent damping property. As a base vibration is transferred to the diaphragm through a substrate layer, mechanical damping in the substrate layer has a negative effect on the vibration damping and vocal sensing performance. Parylene has a low loss factors ($\tan \delta \leq 0.05$; *Polymer Testing* **2016**, 53, 89-97 and *ASME International Mechanical Engineering Congress and Exposition* **2005**, 279-283), which is similar with that of epoxy ($\tan \delta: 0.03 \sim 0.04$; *Polymers and Polymer Composites* **2001**, 9, 423-426), and much lower than that of other elastomers such as PDMS and SBS ($\tan \delta \geq 0.2$; *Journal of Polymer Science Part B: Polymer Physics* **2016**, 54, 747-751 and *Polymers* **2018**, 10, 400).

To attach the sensor device completely onto the neck skin, we applied a bio-compatible adhesive (LP-001, ABLE C&C Co., Ltd). In addition, an ultrathin and transparent medical tape (TegadermTM Film 1622W, 3M) was attached over the edge of the sensor and the metal lines connected to the interface circuit. Only a minimal amount of adhesive (~ 10 mg) was used to attach the device stably so that the adhesive does not affect the sensor operation. We note that the degree of conformal attachment is related to the performance; as shown in Figure 4b, the device exhibits approximately 10% less sensitivity on human neck skins compared to its original vibration sensitivity. Based on the discussions above, we have added the reason for the use of parylene and the information for the device attachment on the neck skin in methods section.

- Revised main text, page 18, line 423-425

“Parylene was used as a polymer substrate due to excellent mechanical stability, bendability, thickness uniformity and low damping property^{46,47}.”

- Added references

(46) Bae, G. Y. et al. Pressure/Temperature Sensing Bimodal Electronic Skin with StimulusDiscriminability and Linear Sensitivity. *Adv. Mater.* **30**, 1803388 (2018).

(47) Chindam, C. et al. Temperature-dependent dynamic moduli of Parylene-C columnar microfibrinous thin films. *Polymer Testing* **53**, 89-97 (2016).

- Revised main text, page 20, line 467-472

“**Attachment of the device on the neck skin.** To attach the sensor device completely onto the neck skin, we applied a bio-compatible adhesive (LP-001, ABLE C&C Co., Ltd). In addition, an ultrathin and transparent medical tape (TegadermTM Film 1622W, 3M) was

attached over the edge of the sensor and the metal lines connected to the interface circuit. Only a minimal amount of adhesive (~ 10 mg) was used to attach the device stably so that the adhesive does not affect the sensor operation.”

Comment 5: In Figure 3, can authors compare the frequency spectrum of the original sound and the spectrum detected by the device?

Response 5: As the reviewer recommended, it is possible to compare the frequency spectrum detected by the sensor and the spectrum of the original sound file. However, in our experiment system, the frequency spectrum of the sound source is different from that of the input of the sensor because the vibration speaker (Newadin Technology, VBT-001) has a non-flat frequency response of the output vibration to the same input voltage, as shown in Figure R2. This characteristic is typical for most vibration speakers. Therefore, we compared the output of the sensor with a certificated commercial accelerometer (PCB Piezotronics, 352C33), as shown in Figure 3c. Because the accelerometer is certificated with a constant sensitivity (0.1 g/V) for the frequency range from 0.1 Hz to 10 kHz, the actual vibrations from the vibration speaker can be quantitatively measured by the accelerometer.

Figure R2. Frequency response data of the vibration speaker (Newadin Technology, VBT-001). The acceleration of output vibration was obtained for the frequencies including the frequency range in Figure 3.

Comment 6: Authors demonstrated a skin-attachable voice recognition sensor. For the broader applications, it is recommended that authors provide the performance of their sensor for the detection of acoustic sounds over the broader frequency range (4-12 kHz) than the vocal frequency range (below 3400 Hz).

Response 6: Our device exhibits a flat frequency response over the frequency range 80 ~ 3400 Hz, which includes the standard telephony bandwidth (300 ~ 3400 Hz) and the

fundamental voice frequency range (80 ~ 250 Hz; *Principles of Voice Production* (Prentice Hall, 1994)). Therefore, we believe that the device exhibits enough performance in detecting human voice for telecommunication and voice recognition applications. As the reviewer mentioned, human voice may have broader spectrum with several harmonics, depending on the vocal quality and gender. For this reason, the state-of-the-art recent audio technology expands the target frequency range up to 7000 Hz (HD voice). However, the purchased commercial vibration speaker exhibits too small and unstable output vibration for the frequency range beyond 4000 Hz, then we could not prove the sensing performance of our device for higher frequencies than the standard telephony bandwidth. We think measurements at such high frequencies are beyond the scope of this manuscript and plan to develop them for ultrasonic vibration sensors in the future work.

Reviewer #2

Overall Comment: The manuscript reports an interesting work with clear writing. However, the novelty of the work seems not particularly strong. It seems that work is just a finding about a good voice sensor and the main contribution of this work is in the use of fully-crosslinked thin polymer and hole-patterned structure as diaphragm. There is little new knowledge developed. I would suggest the authors clearly state the novelty and major contributions of this work.

Response: Thank the reviewer for the critical suggestions and invaluable comments. We revised the manuscript according to the reviewer's comments.

- The novelty and contributions of our research

- 1) We thoroughly investigated the linear correlation between the skin accelerations and human voice pressures. On the basis of the systematic study, we suggest for the first time that flexible and skin-attachable sensor can measure human voice quantitatively by detecting the skin acceleration.
- 2) It is the first time to demonstrate a flexible skin-attachable device (#1) that can perceive human voices quantitatively by detecting the neck skin vibration, as shown in Table R1; so far, all previously reported flexible and skin-attachable sensors (#3-9) have not satisfied the essential requirements of the microphone: flat frequency response and high/linear sensitivity. For high sensing performance, we firstly introduced polymers with low stiffness and low damping properties to sophisticated MEMS structure.

	Journal / Works	Flat frequency response in voice band(80~3400 Hz)	Sensitivity (for 35~70 dB _{SPL})	Linearity (for 35~70 dB _{SPL})
#1	Our work	O	5.5 V/Pa	O
#2	Commercial microphones*	O	0.001~ 1V/Pa	O
#3	Throat microphones (IEEE J. Transl. Eng. Health Med. 3, 1-10 (2015))	O	0.005 V/Pa	
#4	Sci. Adv. 4, eaas8772 (2018)	X	< 0.12V/Pa	X
#5	ACS Nano 9, 4236 (2015)	X	< 2 V/Pa	O
#6	Nat. Commun. 6, 6269 (2015)	X	0.25 V/Pa	O
#7	Energy Environ. Sci. 6, 169-175 (2013)	X	0.1 V/Pa	
#8	Adv. Mater. 27, 1316-1326 (2015)	X	0.051 V/Pa	O
#9	Sci. Adv. 1, e1500661 (2015)	X	0.04 V/Pa	

Table R1. Performance comparison of our work, and previously suggested skin-attachable sensors and commercial microphones.

* Martin, D. T. *Design, fabrication, and characterization of a MEMS dual-backplate capacitive microphone.* Vol. 69 (2007).

- 3) To the best of our knowledge, our device is the only sound/vibration sensor that can be applicable to commercial vocal healthcare monitoring, which requires quantitative voice analysis in noisy work places, as we successfully demonstrated in the manuscript.

Based on the discussions above, we have explained the novelty of our research in more detail in the introduction as follows. The changes are colored in red.

- Revised main text, page 4-5, line 68-95

“In this study, we suggest a novel methodology for the flexible and skin-attachable sensor to satisfy the essential requirements. First, we examined the neck skin vibrations that arise in human speech and ascertained that there is a linear relationship between the skin acceleration and the voice pressure. This finding suggests that the skin acceleration is an appropriate sensing parameter to measure human voice quantitatively. Then, we fabricated an ultrathin and conformable electronic skin by introducing polymers with low damping properties to sophisticated capacitive diaphragm structures ... The device also exhibits superior vibrational sensitivity of 270 mV/g for the range of neck skin vibrations, ... Thus, our device can transduce human voice into electrical voltages with a flat frequency response and a high/linear sensitivity of 5.5 V/Pa. The device is the first flexible and skin-attachable sensor that can perceive human voices quantitatively by detecting the neck skin vibration. In addition, our device detects the human voice clearly in the presence of ambient noise or a mouth mask. The device exhibits not only high sensing performance compared to commercial microphones, but also skin-conformity and noise cancelling functionality even in poor acoustic environments. With these merits, we successfully demonstrated the use of this device in several voice-recognition applications, namely voice authentication, voice remote control systems, and the healthcare monitoring of vocal cords.

Comment 1: There are lots of research works on using variable capacitors to detect vibration and sound, as well as for energy generation. Please give a systemic summary about the previous arts in variable capacitor for vibration sensors and sound detection.

Response 1: As the reviewer commented, we have prepared a systemic summary about previous studies, and the main text has been revised as follows.

- Revised main text, page 3, line 45-66

“Microphones have been developed to detect human voice accurately, and applied in various electronic devices. Especially capacitive microphones, which typically exhibit higher sensitivity and lower noise level than other types of sensors³, have been widely studied. The performance was steadily enhanced by adopting new electret materials⁴ and a variety of structures such as corrugated diaphragm⁵, hole-patterned backplate⁶, and dual backplate⁷, which results in a successful commercialization. ... However, previously reported skin-attachable sensors did not satisfy several essential requirements of microphone: flat frequency response, high sensitivity and linearity of sensitivity. ... Especially, the sensors made of viscoelastic polymers^{10-13,17-19} involving beta transition such as movements of side groups and long molecular chains²³, exhibited considerable damping effects. In addition, such

sensors are not capable of measuring a voice pressure accurately^{8,12}, because a quantitative correlation between neck skin vibration and voice pressure was not fully understood. In addition, such sensors exhibit insufficient and non-linear sensitivity to distinguish subtle differences in voice pressure^{8-11,14,15,18-21}.”

- Added references

- (3) Martin, D. T. et al. A micromachined dual-backplate capacitive microphone for aeroacoustic measurements. *J. Microelectromech. Syst.* 16, 1289-1302 (2007).
- (4) Goel, M. Electret sensors, filters and MEMS devices: New challenges in materials research. *Curr. Sci.* **85**, 443-453 (2003).
- (5) Scheeper, P. R., Olthuis, W. & Bergveld, P. The design, fabrication, and testing of corrugated silicon nitride diaphragms. *J. Microelectromech. Syst.* **3**, 36-42 (1994).
- (6) Scheeper, P., Van der Donk, A., Olthuis, W. & Bergveld, P. Fabrication of silicon condenser microphones using single wafer technology. *J. Microelectromech. Syst.* **1**, 147-154 (1992).
- (7) Bay, J., Hansen, O. & Bouwstra, S. Micromachined double backplate differential capacitive microphone. *J. Micromech. Microeng.* **9**, 30 (1999).

Comment 2: The basic preparation and test methods should be presented in the main text.

Response 2: As the reviewer advised, basic preparation and methods for the neck skin vibration characterization and the sensor test, which were mainly explained in Supplementary Notes, have been transferred to the main text and described in more detail.

- Revised main text, page 5-6, line 101-116

“We measured voice pressures and neck skin vibrations simultaneously while a person spoke at different volume levels with fundamental voice frequencies, which are the largest among numerous harmonics of human voice and represent the voice²⁵. We chose the three frequencies of 100, 150, 200 Hz in the fundamental voice frequency range (80 to 255 Hz) with even intervals.

Voice pressure and neck skin vibration were simultaneously measured while a person spoke (Supplementary Fig. 1a). The voice pressure was measured using a reference microphone placed 1 m in front of the mouth (Brüel & Kjaer, type 4192), and the measured time-domain voltage signal was converted into a frequency domain by a signal analyzer (Stanford Research Systems, SR785). Neck-skin vibration was measured accurately using a laser Doppler vibrometer (LDV; Polytec, OFV-5000). The velocity of neck vibration was measured as a voltage in the time-domain signal, and then converted to a frequency spectrum by a signal analyzer. The amplitude of the fundamental frequency was converted to three vibration parameters: displacement, velocity and acceleration. Finally, we analyzed the correlation between voice pressure and corresponding skin vibration (Supplementary Fig. 1c-

f).”

- Added reference

(25) Redford, M. A. *The handbook of speech production*. (John Wiley & Sons, 2015).

- Revised main text, page 8, line 171-177

“The frequency response sensitivity was defined as the output voltage of the device relative to that of the reference accelerometer (PCB, 352C33), which has a constant sensitivity of 100 mV/g from 10 Hz to 10 kHz (Supplementary Fig. 7). An input vibration was precisely generated by using an electromagnetic vibration speaker (Newadin Technology, VBT-001). To remove any undesirable electromagnetic coupling with the speaker, the device was kept in an aluminum shielding box.”

Comment 3: The authors attributed the excellent flat frequency response of the devices to role of the fully crosslinked polymer diaphragm with a hole patterned structure in reducing damping. However, they fail to explain why.

Response 3: As the reviewer commented, our explanation on the effects of the material/structure damping on the frequency response may be unclear in the previous manuscript. Therefore, we have further strengthened our logic by mentioning how to lower the materials damping in terms of beta transition such as the movements of side groups and long molecular chains of the used polymers. In addition, we have modified a few misleading sentences in the paragraph for the structural damping. We have revised the corresponding paragraphs as follows.

- Revised main text, page 8-9, line 180-199

“The diaphragm material should maintain low loss factors ($\tan \delta$) over the frequency range 80 ~ 3400 Hz to reduce mechanical damping. We utilized a fully crosslinked epoxy resin (SU-8) as diaphragm material. The monomer of the epoxy resin is based on four bisphenol-A units, which exhibit a distinct beta transition at ≈ -100 °C²³ and a glass transition over 150 °C, thereby having low $\tan \delta$ at room temperature. In addition, we further reduced the $\tan \delta$ of the polymer by fully crosslinking the epoxy binding sites during a hard baking process at a temperature of 240 °C. The fully crosslinked network structure prevents the oscillation and conformation flip of the phenyl rings in bisphenol-A^{29,30}. Therefore, our fully crosslinked SU-8 exhibits a wider bandwidth of flat frequency response (Fig. 2a, blue) than poly(methyl methacrylate) (PMMA) with a similar mass density and stiffness but higher $\tan \delta$ due to a rotating motion of the ester groups²³ (Fig. 2a, purple).

According to a theory of squeezed-film air damping³¹, an air film under a diaphragm, isolated from ambient air, has unfavorable effects on the diaphragm frequency response. As the volume of air film is decreased and the ratio of lateral dimension to height in the air film

is increased over twenty times, the structural damping becomes more significant. For this reason, the polymer diaphragm without holes has a flat frequency response up to only 1300 Hz in spite of the reduced material damping (Fig. 2a, blue). To decrease the structural damping effects, we patterned eight holes in the diaphragm to ventilate an air beneath the diaphragm to the ambient air, which results in improved flat frequency response up to 3,500 Hz (Fig. 2a, olive).”

- Added references

(23) Monnerie, L., Lauprêtre, F. & Halary, J. L. Investigation of Solid-State Transitions in Linear and Crosslinked Amorphous Polymers. *Adv. Polym. Sci.* **18**, 35–213 (2005).

(29) Schmid, S. & Hierold, C. Damping mechanisms of single-clamped and prestressed double-clamped resonant polymer microbeams. *J. Appl. Phys* **104**, 093516 (2008).

(30) Chung, S. & Park, S. Effects of temperature on mechanical properties of SU-8 photoresist material. *J. Mech. Sci. Technol.* **27**, 2701-2707 (2013)

Comment 4: Page 7 the sentences “...Most previous flexible and skin-attachable sensors consisted of viscoelastic... material damping limits sensitivity as the input frequency increases.” are background, they should be moved to the introduction section.

Response 4: As the reviewer advised, the sentence above has been moved to the introduction.

- Revised main text, page 3, line 60-62

“Especially, the sensors made of viscoelastic polymers^{10-13,17-19} involving beta transition such as movements of side groups and long molecular chains²³, exhibited considerable damping effects.”

Reviewer #3

Overall Comment: This work presents an ultrathin device for voice sensing and recognition via acceleration measurement on human cricoid cartilage. Applications including voice authentication, speech recognition and voice dosimetry for vocal health monitoring are demonstrated. Taking advantage of specially designed structures, the device is described to show superior performances comparing to the commercial accelerometer and microphones under several application scenarios. However, several significant details including the wearability of the device, vocal pressure-skin acceleration relation characterization, and the performance characterization are not described in a sufficiently clear manner. The manuscript would be stronger if the author can address the following questions and supplement that information to the current content.

Response: We would like to thank the critical suggestions and invaluable comments. We revised the manuscript according to the reviewer's comments.

Comment 1: The author mentioned that the human neck skin possesses a curvature of approximately 16 m^{-1} . However, the neck skin should also have a curvature distribution, and the curvature near cricoid cartilage should be naturally higher than other parts. Furthermore, varying with subjects, the local surface near cricoid cartilage may not be developable and could be dynamic during speaking and motion. Can the device guarantee the conformity under all these situations?

Response 1: We agree with the reviewer's comment that the neck skin has a curvature distribution and the curvature near the cricoid cartilage is higher than the other neck area. However, as shown in Figure 1c, the ultrathin device can be conformably attached on a curved surface with a small bending radius of only 2.5 mm, allowing conformal contact to the neck skin around the cricoid cartilage. As the reviewer mentioned, the local surface near cricoid cartilage can be dynamic during speaking and motion. To maintain the conformal contact of the device even in these situations, we applied a bio-compatible adhesive (LP-001, ABLE C&C Co., Ltd). Only a minimal amount of adhesive ($\sim 10 \text{ mg}$) was used to attach the device stably so that the adhesive does not affect the sensor operation. In addition, an ultrathin and transparent medical tape (TegadermTM Film 1622W, 3M) was attached over the edge of the sensor and the metal lines connected to the interface circuit. We have confirmed that the device is conformably contacted and capable of detecting human voice clearly even under dynamic situations such as shaking the head back and forth, as shown in Figure R3.

Figure R3. Output waveform and frequency spectrum measured by our device attached on the neck skin while the user phonated “Siyoung log-in” with leaning head back (a) and bending forth (b). Photographs in (a) and (b) show our device attached on the neck skin when the user shakes the head.

Based on the discussions above, we have added the information for the device attachment on the neck skin into the method section.

- Revised main text, page 20, line 467-472

“Attachment of the device on the neck skin. To attach the sensor device completely onto the neck skin, we applied a bio-compatible adhesive (LP-001, ABLE C&C Co., Ltd). In addition, an ultrathin and transparent medical tape (TegadermTM Film 1622W, 3M) was attached over the edge of the sensor and the metal lines connected to the interface circuit. Only a minimal amount of adhesive (~ 10 mg) was used to attach the device stably so that the adhesive does not affect the sensor operation.”

We also have added the related experimental results to the main text and Supplementary Figure 17.

- Revised main text, page 13, line 303-305

“In addition, our device is capable of detecting human voice clearly even under dynamic situations such as shaking the head back and forth (Supplementary Fig. 17).”

- Revised Supplementary Information, page 30

Figure R3 → Supplementary Figure 17

Comment 2: How is the device fixed on the human neck? Is there any adhesive material on the substrate to prevent the device from delamination? How long can a subject wear the device? Will there be skin irritation or itch after a certain period of wearing? The authors are suggested to add information on those aspects.

Response 2: To maintain a conformal and stable contact of the device even, we applied a bio-compatible adhesive (LP-001, ABLE C&C Co.). Only a minimal amount of adhesive (~ 10 mg) was used to attach the device stably so that the adhesive does not affect the sensor operation. In addition, an ultrathin and transparent medical tape (Tegaderm Film™ 1622W, 3M) was attached over the edge of the sensor and the metal lines connected to the interface circuit. The bio-compatible adhesive is one of the cosmetics ingredients and can be easily removed by a make-up remover. The medical tape is widely utilized for skin attachment. Therefore, we did not experience a skin irritation or itch after three hours of wearing as shown in Figure R4.

Figure R4. Photograph showing the neck skin after three hours of wearing our device. This figure has been added as Supplementary Figure 20.

Based on the discussions above, we have added bio-compatibility information of the device attachment on the neck skin to the methods section and Supplementary Figure 20.

- Revised main text, page 20, line 472-475

“The bio-compatible adhesive is one of the cosmetics ingredients and can be easily removed by a make-up remover. The medical tape is widely utilized for skin attachment. Therefore, we did not experience a skin irritation or itch after three hours of wearing (Supplementary Fig. 20)”

- Revised Supplementary Information, page 33

Figure R4 → Supplementary Figure 20

Comment 3: In supplementary note 1 and supplementary figure 1, the authors mentioned a laser Doppler instrument associated with a tiny laser mirror was used for neck skin vibration measurement. The voice was also recorded by a microphone simultaneously to correlate with the vibration measured.

Comment 3-(1): What is the relative size of the mirror comparing to the cricoid cartilage? If the mirror is smaller, will its position influence the result?

Response 3-(1): The size of the mirror was approximately 1.5 cm², which is slightly smaller than the area of the neck skin on cricoid cartilage. The acceleration of skin vibration can be different depending on the position of the mirror, as shown in our experiment result (Supplementary Figure 22). Therefore, we set up an accurate position of the mirror above the frontal part of cricoid cartilage based on a previous vocal study (*Engineering in Medicine and Biology Society* 2009, 4453-4456) for the mapping of neck skin vibrations during vocalized speech (Supplementary Note 5). In addition, we adjusted the angle of the mirror to be perpendicular to the source of the laser Doppler vibrometer for accurate and stable focus of the laser source. Based on the discussions above, we have added this explanation to the methods section.

- Revised main text, page 19, line 453-458

“A tiny piece of laser mirror sheet was softly attached onto the skin above the cricoid cartilage, which vibrates when a human speaks (Supplementary Fig. 1b). The exact position of the mirror was defined, based on a previous vocal study⁵⁰ for the mapping of neck surface vibrations during human speech. We adjusted the angle of the mirror to be perpendicular to the source of the laser Doppler vibrometer for accurate and stable focus of the laser source.”

- Added reference

(50) Nolan, M. et al. Accelerometer based measurement for the mapping of neck surface vibrations during vocalized speech. 31st Annual International Conference of IEEE Engineering in Medicine and Biology Society 4453-4456 (2009).

Comment 3-(2): In supplementary figure 1c, why are the three certain frequency values (100, 150 and 200 Hz) chosen for analysis?

Response 3-(2): Fundamental voice frequency is the largest among numerous harmonics of human voice, and then represents the voice. Therefore, we chose the three frequencies of 100, 150, 200 Hz in the fundamental voice frequency range (80 to 255 Hz) with even intervals. We have added this explanation to the main text.

- Revised main text, page 5, line 101-105

“We measured voice pressures and neck skin vibrations simultaneously while a person spoke at different volume levels with fundamental voice frequencies, which are the largest among numerous harmonics of human voice and represent the voice²⁵. We chose the three frequencies of 100, 150, 200 Hz in the fundamental voice frequency range (80 to 255 Hz) with even intervals.”

- Added reference

(25) Redford, M. A. *The handbook of speech production*. (John Wiley & Sons, 2015).

Comment 3-(3): What are the uncertainty levels of both measurement methods?

Response 3-(3): The reference microphone (Bruel & Kjaer, microphone type 4192) has an uncertainty level of approximately $\pm 12.2\%$ (1 dB) at the test frequency range. The laser Doppler vibrometer (Polytec, OFV-5000) exhibits a resolution of 0.02 $\mu\text{m/s}$ with VD-09 decoder. The signal analyzer (Stanford Research Systems, SR785; sample acquisition time: > 200 ms), converting time-domain voltage signal to the frequency domain, has an uncertainty of approximately $\pm 2.3\%$ (0.2 dB) for amplitude measurement. We have added this information to the methods section.

- Revised main text, page 19, line 449-453

“**Characterization of neck skin vibrations.** The reference microphone has an uncertainty level of approximately $\pm 12.2\%$ (1 dB) at the tested frequency range. The laser Doppler vibrometer exhibits a resolution of 0.02 $\mu\text{m/s}$ with VD-09 decoder. The signal analyzer (sample acquisition time: > 200 ms), converting time-domain voltage signal to the frequency domain, has an uncertainty of approximately $\pm 2.3\%$ (0.2 dB) for amplitude measurement.”

Comment 4: In the manuscript, spectrums are frequently used for device performance comparison, and the author mentioned that the fast Fourier transform (FFT) was used for joint time-frequency analysis.

Comment 4-(1): What is the sampling frequency of the time-domain signal? What are the parameters, including window length, overlap, and window function used for FFT?

Response 4-(1): The sampling frequency of the time-domain signal was 10 kHz, window length was 0.2 second (data length is 2,000), and overlap was 0.05 sec. No window function was not used for FFT. We have added this information to the methods section.

- Revised main text, page 20-21, line 477-479

“...the frequency domain was analyzed with FFT by using Matlab (MathWorks, R2016a). The sampling frequency of the time-domain signal was 10 kHz, window length was 0.2 second (data length is 2,000), overlap was 0.05 sec, and no window function was not used during FFT.”

Comment 4-(2): The author mentioned the device exhibited a sufficient frequency resolution and short response time as described in supplementary figure 14, which is not clear to the reviewer. To the reviewer’s knowledge, the parameters of FFT will significantly influence the temporal and frequency resolution results if they are not chosen properly. A further detailed and systematic study is required to characterize the temporal and frequency resolution of the device.

Response 4-(2): We agree with the reviewer’s comment that the terms ‘sufficient frequency resolution’ and ‘short response time’ are not clear. ‘Pachelbel’s Canon in C’ (Fig. 3) was chosen to demonstrate that our sensor is capable of detecting input vibrations without distortion, because this music consists of two tones of adjacent frequencies and fast beats of a sixteenth note in 80 beats per minute. ‘Knocking on heaven’s door’ (Supplementary Fig. 14 in the original manuscript) was chosen to demonstrate that our sensor is able to process the complex frequency spectrum arising from various instruments and voices. For the measurement of the music performance, we set the frequency resolution and the window length to be 5 Hz and 0.2 second, respectively, in the sampling frequency of 10 kHz. These values have trade-off relationship of ‘frequency resolution (5 Hz) × data length (2000) = sampling frequency (10 kHz)’. Therefore, the values were set, considering not only frequency difference of human voices and music tones, but also beats of music. As a result of demonstration, we confirmed that the device can clearly distinguish every music notes, based on the peaks of frequency spectrum obtained by our device (Figure 3b). Based on the discussions above, we have revised the previous misleading sentence as follows.

- Revised main text, page 12, line 265-269

“... Moreover, the device could distinguish the performed music sheets (Fig. 3a), consisting of two tones of adjacent frequencies and a sixteenth note in 80 beats per minute, as shown in the peaks in the frequency spectrum (Fig. 3b). The device can detect complicated frequency spectrum from various instruments and voices as well (Supplementary Fig. 16a).”

Comment 4-(3): There are two different plotting styles of the spectrums (figure 3b,c and supplementary figure 14 belong to one, figure 5 and supplementary figure 15 belong to the other one). The reviewer suggests that the author should unify these two plotting styles.

Response 4-(3): We agree with the reviewer’s comment that it is better to have the two plots in the same styles. We have modified Figure 3b,c and Supplementary Figure 14 with the plot styles of Figure 5 and Supplementary Figure 15.

- Figure 5b

- Revised Figure 3b,c

- Revised Supplementary Figure 14 (Supplementary Figure 16 in the revised version)

Comment 5: In figure 3 and supplementary figure 7, the author used a commercial speaker to characterize the device performance compared with a commercial accelerometer.

Comment 5-(1): In this experiment, the author horizontally laminated the device on the speaker. The orientation of the device is different when it is worn on the subject's neck. Will the gravity affect the performance of the device under these two different situations?

Response 5-(1): The orientation of the device could affect slightly the initial deflection of the suspended diaphragm. The diaphragm of the device laid horizontally can be more deflected compared to the device laid vertically, considering the mass and stiffness of the diaphragm are 1.16×10^{-10} kg and 37.71 N/m, respectively. However, the output voltage of the device is not correlated to the initial deflection of the diaphragm, but correlated to the distance change between the top and bottom electrodes. Therefore, the orientation of the device does not have a significant effect on the device performance.

Comments 5-(2): The author chose two periods of music, “Pachelbel’s Canon in C” and “knocking on the heaven’s door” for performance characterization. However, the total frequency range of these two audios showed in the manuscript only covers 130 ~ 800 Hz, which is still far from the whole audible frequency range. The reviewer suggests that the author should conduct experiments in broader frequency ranges.

Response 5-(2): We agree with the reviewer’s comment that the range of the spectrum is less than the whole audible frequency range. We showed a fundamental frequency range corresponding to music note (Figure 3a) to facilitate comparison between our device and the commercial accelerometer. As the reviewer advised, we have added the frequency spectra of “Pachelbel’s Canon in C” with frequency range from 80 Hz to 2000 Hz, which is obtained by our device and commercial accelerometer into Supplementary Information, as shown in Figure R5. In addition, we have modified the frequency data of “knocking on the heaven’s door” (Supplementary Figure 14) with the frequency range from 80 Hz to 2000 Hz, as shown in Figure R6.

- Revised main text, page 11, line 255-256

“... ‘Pachelbel’s Canon in C’ (Fig. 3; see Supplementary Fig. 15 for the analysis with broader frequency range)...”

- Revised Supplementary Information, page 28-29

Figure R5 → Supplementary Figure 15

Figure R6 → Supplementary Figure 16

Figure R5. Comparative sound recognition test with commercial accelerometer. For Pachelbel's Canon (a) was played by a vibration speaker, frequency spectrum from our device (b) and a commercial accelerometer (sensitivity: 100 mV/g for 10 ~ 10,000 Hz) (c) as a function of time. Frequency range of frequency spectra is broader (80 Hz to 2000 Hz) than that of frequency spectra obtained by our device (Fig. 3a) and commercial accelerometer (Fig. 3b). Figure 3 showed only fundamental frequency range corresponding to music note (Figure 3a) to facilitate comparison between our device and the commercial accelerometer. This figure has been added as Supplementary Figure 15.

Figure R6. Comparative sound recognition test with commercial accelerometer. Comparative experiment of the device for recoding a part of music *'knocking on the heaven's door'* (Bob dylan). Waveform and frequency spectrum of output data using our device **(a)** and the reference accelerometer (PCB, 352C33) **(b)**. This figure has been added as Supplementary Figure 16.

Comment 5-(3): For the comparison to the commercial accelerometer, the author was more focused on the sensitivity differences. The reviewer suggests that the author should also compare the spectral noise level with that of the commercial accelerometer, which is also a significant characteristic.

Response 5-(3): We agree with the review’s comment that the spectral noise level is also a significant characteristic and should be compared. We obtained the noise power spectral density of our device and the commercial accelerometer (PCB Piezotronics, 352C33) for voice frequency range (80 ~ 3400 Hz) by connecting the two devices to a signal analyzer (Stanford Research Systems, SR785). As shown in Figure R7, our device has approximately 10 times higher spectral noise level than that of the commercial accelerometer. This result is attributed to the difference of electromagnetic shielding and electric wiring design. We believe that well-designed wiring and packaging technique, related to electromagnetic noise, is beyond the scope of current study. Nonetheless, our device can be utilized as a voice-recognition device with outstanding reliability, because of the high signal-to-noise ratio of more than 10 dB for a vibration of 0.02 g (Supplementary Fig. 12), which is the smallest neck-skin vibration in human speech.

Figure R7. Noise power spectral density of our device (a) and the commercial accelerometer (PCB Piezotronics, 352C33) (b) for voice frequency range (80 ~ 3400 Hz). The data was obtained by connecting the two devices to a signal analyzer (Stanford Research Systems, SR785). This figure has been added as Supplementary Figure 12.

Based on the discussions above, we have added the information on the spectral noise level of our device to the main text and Supplementary Figure 12.

- Revised main text, page 10-11, line 236-241

“Our device has approximately 10 times higher spectral noise level than that of the commercial accelerometer (Supplementary Fig. 12) due to the difference of electromagnetic shielding and electric wiring design. Nonetheless, our device exhibits a high signal-to-noise ratio of more than 10 dB for a vibration of 0.02 g (Supplementary Fig. 13), which is the smallest neck-skin vibration arising in human speech.”

- Revised Supplementary Information, page 25

Figure R7 → Supplementary Figure 12

Comment 6: In figure 2a, the author showed the sensitivity-frequency relation of devices with holes made by low loss factors. A measurement of the device with high loss factors and holes should also be added to facilitate the comparison.

Response 6: The purpose of analysis in Figure 2a is to show that both of material and structural damping effects should be reduced for the wider bandwidth of flat frequency response in flexible and skin-attachable sensors. For this reason, the material damping effect was studied by comparing SU-8 and PMMA, which have similar mechanical properties such as mass density and stiffness, but different loss factors. In addition, the structural damping effect was analyzed by comparing the SU-8 diaphragms with and without air holes; these two sample sets were made by the same fabricating process and of the same materials. Therefore, we believe that the two comparative experiments not only demonstrated the device with a flat frequency response over the frequency range of 80 ~ 3400 Hz, but also fully explained the general methodology to reduce material and structural damping for flexible and skin-attachable sensors.

We agree with the reviewer’s comment that an additional measurement could facilitate the comparison. The device with high loss factor and air holes would have narrower flat frequency range than the device with low loss factor and air holes, and wider range than the device with high loss factor and without air hole. The device with high loss factor and air holes would be compared to the device with low loss factor and without air-hole to compare material and structural damping effects on the dominance of the frequency range. However, this comparison is valid only for the current device system and lacks of generality for flexible and skin-attachable sensors with various materials and structures. Therefore, we focused on conducting two controlled experiments, showing the effects of loss factors and diaphragm structure respectively.

REVIEWERS' COMMENTS:

Reviewer #1 (Remarks to the Author):

Authors fully addressed all the comments raised by the reviewers. All the claims were supported by the results and explanations.

Reviewer #2 (Remarks to the Author):

The manuscript after revision looks good. It addresses all my concerns. Acceptance is recommended in the current form.

Reviewer #3 (Remarks to the Author):

In this revised version of the manuscript, the authors have written a comprehensive response letter, answering the questions one-by-one. The reviewer feels that the authors have successfully enhanced their content and understanding of the device performance, wearability, and data decoding protocols. And most of that supplemented knowledge was integrated into the revised manuscript and SI. The reviewer now recommended this work to be published in Nature Communication.

RESPONSE TO THE REVIEWERS' COMMENTS

(Manuscript ID: NCOMMS-18-33828A)

Reviewer #1

Overall Comment: Authors fully addressed all the comments raised by the reviewers. All the claims were supported by the results and explanations.

Response: We are grateful to the reviewer for taking time to review our manuscript and for positive assessment of our work.

Reviewer #2

Overall Comment: The manuscript after revision looks good. It addresses all my concerns. Acceptance is recommended in the current form.

Response: We are grateful to the reviewer for taking time to review our manuscript and for positive assessment of our work.

Reviewer #3

Overall Comment: In this revised version of the manuscript, the authors have written a comprehensive response letter, answering the questions one-by-one. The reviewer feels that the authors have successfully enhanced their content and understanding of the device performance, wearability, and data decoding protocols. And most of that supplemented knowledge was integrated into the revised manuscript and SI. The reviewer now recommended this work to be published in Nature Communication.

Response: We are grateful to the reviewer for taking time to review our manuscript and for positive assessment of our work.